# Malaria parasites use a soluble RhopH complex for erythrocyte invasion and an integral form for nutrient uptake

Marc A Schureck[1], Joseph E Darling[2], Alan Merk[2], Jinfeng Shao[1],
Geervani Daggupati[3], Prakash Srinivasan[3], Paul Dominic B Olinares[4],
Michael P Rout[5], Brian T Chait[4], Kurt Wollenberg[6], Sriram Subramaniam[7]*,
Sanjay A Desai[1]*

[1]Laboratory of Malaria and Vector Research, NIAID, National Institutes of Health, Rockville, United States; [2]Laboratory of Cell Biology, Center for Cancer Research, National Cancer Institute, National Institutes of Health, Bethesda, United States; [3]Department of Molecular Microbiology and Immunology, and Johns Hopkins Malaria Institute, Johns Hopkins Bloomberg School of Public Health, Baltimore, United States; [4]Laboratory of Mass Spectrometry and Gaseous Ion Chemistry, The Rockefeller University, New York, United States; [5]Laboratory of Cellular and Structural Biology, The Rockefeller University, New York, United States; [6]Office of Cyber Infrastructure & Computational Biology, National Institute of Allergy and Infectious Diseases, National Institutes of Health, Bethesda, United States; [7]Department of Biochemistry and Molecular Biology, University of British Columbia, Vancouver, Canada

*For correspondence:
Sriram.Subramaniam@ubc.ca (SS);
sdesai@niaid.nih.gov (SAD)

**Abstract** Malaria parasites use the RhopH complex for erythrocyte invasion and channel-mediated nutrient uptake. As the member proteins are unique to Plasmodium spp., how they interact and traffic through subcellular sites to serve these essential functions is unknown. We show that RhopH is synthesized as a soluble complex of CLAG3, RhopH2, and RhopH3 with 1:1:1 stoichiometry. After transfer to a new host cell, the complex crosses a vacuolar membrane surrounding the intracellular parasite and becomes integral to the erythrocyte membrane through a PTEX translocon-dependent process. We present a 2.9 Å single-particle cryo-electron microscopy structure of the trafficking complex, revealing that CLAG3 interacts with the other subunits over large surface areas. This soluble complex is tightly assembled with extensive disulfide bonding and predicted transmembrane helices shielded. We propose a large protein complex stabilized for trafficking but poised for host membrane insertion through large-scale rearrangements, paralleling smaller two-state pore-forming proteins in other organisms.

## Introduction

Malaria parasites evade host immunity by replicating within vertebrate erythrocytes. In humans, the virulent *Plasmodium falciparum* pathogen uses multiple ligands for erythrocyte invasion (*Cowman et al., 2012*) and then remodels its host cell to achieve tissue adherence and nutrient acquisition (*Goldberg and Cowman, 2010*; *Wahlgren et al., 2017*; *Desai, 2014*). Remarkably, a single protein complex, termed RhopH, contributes to each of these activities despite their separate timings and cellular locations (*Gupta et al., 2015*; *Goel et al., 2010*). The three subunits of the RhopH complex, known as CLAG, RhopH2, and RhopH3, are conserved and restricted to Plasmodium spp.; none have significant homology to proteins in other genera (*Kaneko, 2007*), suggesting

**eLife digest** Malaria is an infectious disease caused by the family of *Plasmodium* parasites, which pass between mosquitoes and animals to complete their life cycle. With one bite, mosquitoes can deposit up to one hundred malaria parasites into the human skin, from where they enter the bloodstream. After increasing their numbers in liver cells, the parasites hijack, invade and remodel red blood cells to create a safe space to grow and mature. This includes inserting holes in the membrane of red blood cells to take up nutrients from the bloodstream.

A complex of three tightly bound RhopH proteins plays an important role in these processes. These proteins are unique to malaria parasites, and so far, it has been unclear how they collaborate to perform these specialist roles.

Here, Schureck et al. have purified the RhopH complex from *Plasmodium*-infected human blood to determine its structure and reveal how it moves within an infected red blood cell. Using cryo-electron microscopy to visualise the assembly in fine detail, Schureck et al. showed that the three proteins bind tightly to each other over large areas using multiple anchor points. As the three proteins are produced, they assemble into a complex that remains dissolved and free of parasite membranes until the proteins have been delivered to their target red blood cells. Some hours after delivery, specific sections of the RhopH complex are inserted into the red blood cell membrane to produce pores that allow them to take up nutrients and to grow.

The study of Schureck et al. provides important new insights into how the RhopH complex serves multiple roles during *Plasmodium* infection of human red blood cells. The findings provide a framework for the development of effective antimalarial treatments that target RhopH proteins to block red blood cell invasion and nutrient uptake.

that these proteins and the complex they form evolved to meet the specific demands of blood-stream parasite survival.

While RhopH2 and RhopH3 are single-copy genes in all Plasmodium spp., CLAG proteins are encoded by a multigene family with variable expansion in malaria parasite species infecting humans and other vertebrates including birds, rodents, and primates (*Kaneko et al., 2001*; *Cortés et al., 2007*; *Rovira-Graells et al., 2015*). Each of these subunits is transcribed in mature schizont-infected erythrocytes (*Figure 1A*; *Ling et al., 2004*); during translation, these proteins assemble with unknown stoichiometries into a complex that is packaged into rhoptry organelles (*Ito et al., 2017*). Upon host cell rupture, RhopH3, but not CLAG or RhopH2 subunits, facilitates invasion of the next erythrocyte. Some 18 hr later, CLAG3, a paralog encoded by the parasite chromosome 3, inserts in the host erythrocyte membrane to form the plasmodial surface anion channel (PSAC) for nutrient uptake (*Desai et al., 2000*; *Nguitragool et al., 2011*; *Pillai et al., 2012*); other paralogs may also contribute to PSAC (*Gupta et al., 2020*) or, in the case of CLAG9, to cytoadherence (*Trenholme et al., 2000*; *Goel et al., 2010*; *Nacer et al., 2011*). RhopH2 and RhopH3 also traffic to the host membrane and are required for PSAC activity (*Ito et al., 2017*; *Counihan et al., 2017*). Because these proteins have no homologs in other genera, how they traffic within infected cells and serve these multiple roles is unknown.

Our data reveal essential features of the RhopH complex. We combine mass spectrometry, single-particle cryo-electron microscopy (cryo-EM) and biochemical studies using conditional knockdown of protein export to determine that the RhopH is initially produced as a soluble complex that functions in erythrocyte invasion. The complex remains soluble in extracellular merozoites and, upon completed invasion, is deposited into the parasitophorous vacuole surrounding the intracellular parasite. A protein translocon on the parasitophorous vacuolar membrane, PTEX (*de Koning-Ward et al., 2009*), contributes to RhopH export via an unknown mechanism (*Ito et al., 2017*). Our high-resolution de novo RhopH complex structure and biochemical studies suggest large-scale conformational changes for eventual conversion to an integral form at the host erythrocyte membrane. This conversion is PTEX dependent and enables channel-mediated uptake of host plasma nutrients.

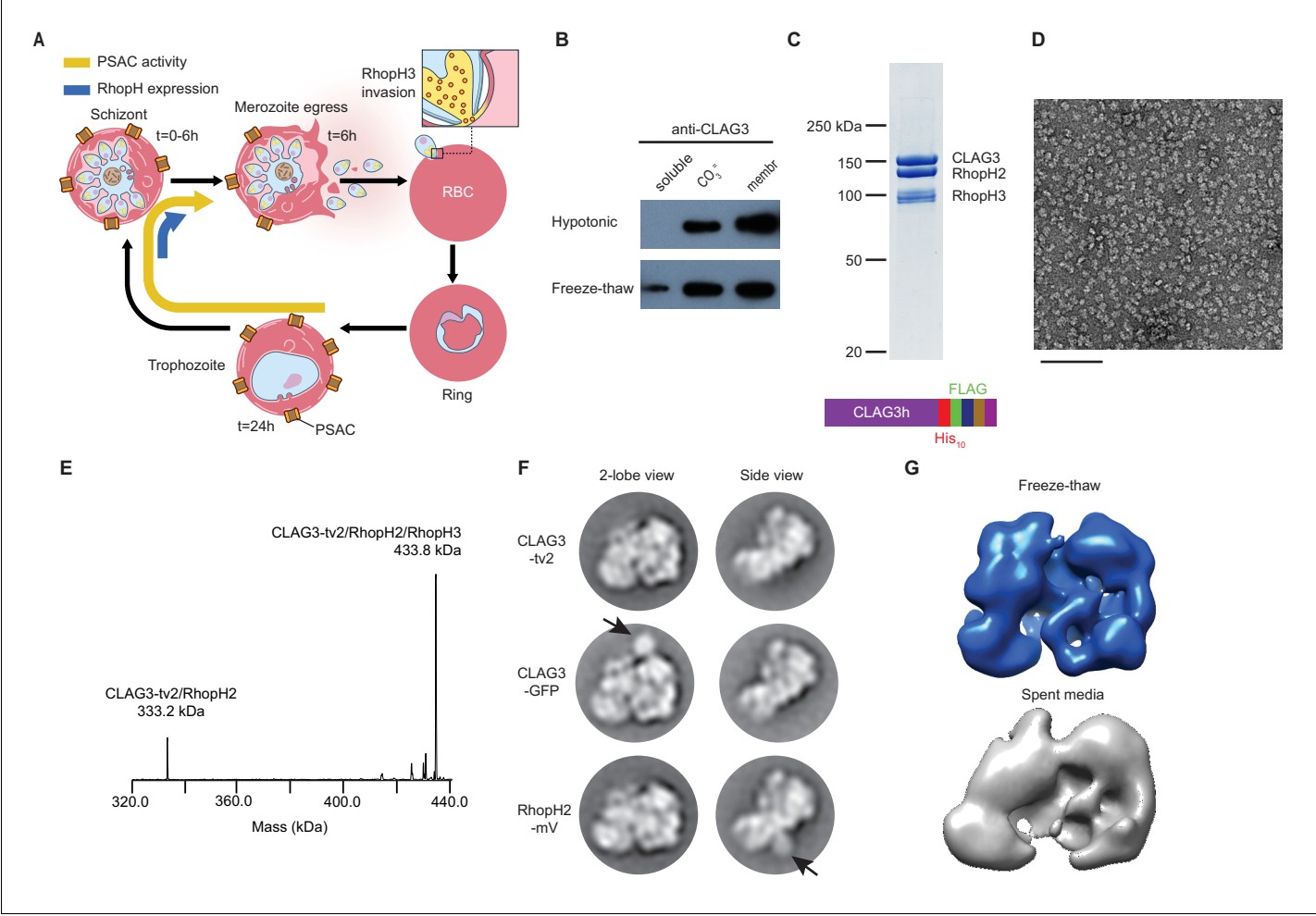

**Figure 1.** A stable, soluble RhopH complex with a 1:1:1 subunit stoichiometry in schizont-infected erythrocytes. (**A**) Schematic showing RhopH complex synthesis in schizonts (t = 0–6 hr), role of RhopH3 in erythrocyte invasion (t = 6 hr), and contribution to plasmodial surface anion channel (PSAC)-mediated nutrient uptake at the host membrane (t = 24–44 hr). (**B**) Immunoblot showing that hypotonic lysis does not release CLAG3, but that alkaline carbonate treatment ($CO_3^=$) and freeze–thaw release distinct pools from membranes (membr). (**C**) Coomassie-stained gel of three RhopH proteins recovered by coimmunoprecipitation after freeze–thaw release. Ribbon at bottom, C-terminal multi-tag strategy used for purification of CLAG3-tv2. (**D**) Negative staining electron microscope image of purified RhopH complexes; scale bar, 100 nm. (**E**) Deconvolved native mass spectrometry (MS) spectrum for endogenous RhopH complexes. (**F**) Negative stain 2D class averages without (top) or with C-terminal tagging with green fluorescent protein (GFP) variants (CLAG3-GFP or RhopH2-mVenus, respectively). GFP-variant density is denoted by black arrows. (**G**) Negative stain 3D reconstructions using freeze–thaw preparation or harvest from spent media without freeze–thaw. Note similar architectures.

The online version of this article includes the following figure supplement(s) for figure 1:

**Figure supplement 1.** Native mass spectrometry (MS) analysis of the endogenous RhopH complex immunoprecipitated via CLAG3-tv2.

**Figure supplement 2.** Subunit modifications and thermostability of the RhopH complex.

## Results

### Freeze–thaw releases a soluble RhopH complex with 1:1:1 subunit stoichiometry

To address these questions, we sought to recover well-behaved RhopH complexes. Alkaline $Na_2CO_3$ extraction but not hypotonic treatment partially released CLAG3 from infected cell membranes (*Figure 1B*, top row), implicating both integral and peripheral membrane pools. We found that simple freeze-thaw also releases some CLAG3 from the peripheral pool (bottom row); although $Na_2CO_3$ extraction releases a larger amount, freeze–thaw is gentler and does not denature many proteins.

Neither of these treatments is expected to release integral membrane proteins. Using multiple C-terminal tags engineered into the single *clag3h* gene of the KC5 line (*Gupta et al., 2018*; CLAG3-tv2; *Figure 1C*, bottom), we effectively harvested this minor fraction from human blood cultures. This CLAG3 remained associated with RhopH2 and RhopH3 (*Figure 1C*) and yielded monodisperse protein complexes in negative stain imaging (*Figure 1D*). Native mass spectrometry (MS) yielded a molecular weight of 433,790 ± 10 Da (*Figure 1E*, *Figure 1—figure supplement 1*), matching the expected mass for a heterotrimeric complex with a 1:1:1 stoichiometry; a 0.6% mass error may reflect post-translational modification and/or proteolytic processing, as reported for RhopH3 (*Ito et al., 2017*). A smaller 333,232 ± 3 Da fraction corresponded to a minor CLAG3–RhopH2 heterodimer. Thus, freeze–thaw permits gentle, detergent-free harvest of this essential complex.

RhopH complexes segregated into 2D classes with two primary views (two-lobe and side views, *Figure 1F*, top row). We next used a green fluorescent protein (GFP)-derivative-tagging approach (*Ciferri et al., 2012*), confirmed integrity of each variant (*Figure 1—figure supplement 2A*), and detected densities reflecting addition of this globular epitope tag. This independently confirmed single copies of each subunit and established an orthogonal arrangement for CLAG3 and RhopH2 (arrows, *Figure 1F*). Three-dimensional reconstruction provided a low-resolution image of the entire complex and established a two-lobed structure (*Figure 1G*). A similar two-lobed structure was obtained for RhopH complexes recovered from spent media without protease inhibitors, detergents, or freeze–thaw, implicating a highly stable complex. Finally, the purified RhopH complex resisted aggregation and unfolding at temperatures above those seen in malaria fevers (*Figure 1—figure supplement 2B,C*). We submit that a thermostable RhopH complex is well-equipped for transit through diverse subcellular environments.

## Structure of the RhopH complex

We next determined the complex's de novo structure using cryo-EM and concentrated protein from sequential coimmunoprecipitation (0.8–2 mg/mL, FLAG and His$_{10}$ tags). Initial analyses with 2D and 3D classifications yielded a two-lobed structure with a 3.3 Å resolution (*Figure 2, Figure 2—figure supplement 1*; *Table 1*); per-particle contrast transfer function (CTF) estimation and motion correction improved overall resolution to 2.9 Å.

Soluble RhopH is a heterotrimeric complex consisting of single CLAG3, RhopH2, and RhopH3 subunits (*Figure 2A*), as predicted above. CLAG3 mediates subunit associations through independent contacts with RhopH2 and RhopH3, which do not directly interact with one another. The visualized complex assumes a 'shallow bowl with a short base' appearance due to an out-of-plane orientation of RhopH2 relative to CLAG3 (*Figure 2B*). On the opposite face, a CLAG3 mid-section protrudes to create a short 'base' that includes a critical amphipathic α-helix proposed to line the PSAC pore at the host membrane, as described below. The bowl's opposite rim is formed by globular α-helices from CLAG3 and RhopH3.

From other angles, an asymmetric two-lobed architecture is apparent, with a well-resolved large lobe that enabled confident de novo model building for CLAG3 and RhopH3 (*Figure 2C*, *Figure 2—figure supplement 1C*). In contrast, the small lobe was initially not well-resolved.

We hypothesized that both lobes have defined structures that undergo relative movement and therefore used multi-body refinement (*Nakane et al., 2018*) to identify rigid but mobile substructures. Assuming two bodies joined by a CLAG3 stem, we refined each lobe separately and improved the small lobe's resolution (*Figure 2—figure supplement 1C*). The small lobe's hammer-shaped ends were now clearly visualized, improving model building from 225 to 513 residues for RhopH2. Excluding their flexible N- and C-terminal tails, ≥90% of CLAG3 and RhopH3 residues were also confidently localized. Multibody refinement also defined the directions and extent of motion between the two lobes (*Figure 2—figure supplement 2*; *Videos 1* and *2*). Interestingly, consideration of protein energy landscapes using normal mode analysis (*Suhre and Sanejouand, 2004*) predicted remarkably similar motions (*Videos 3–7*). Although the biological significance of this mobility is uncertain, conservation of the stem sequence and length in *P. falciparum* CLAG paralogs and among other Plasmodium spp. supports an important role (*Figure 2—figure supplements 3* and *4*; 48% bridge region identity between divergent human *P. falciparum* and *P. vivax* CLAGs).

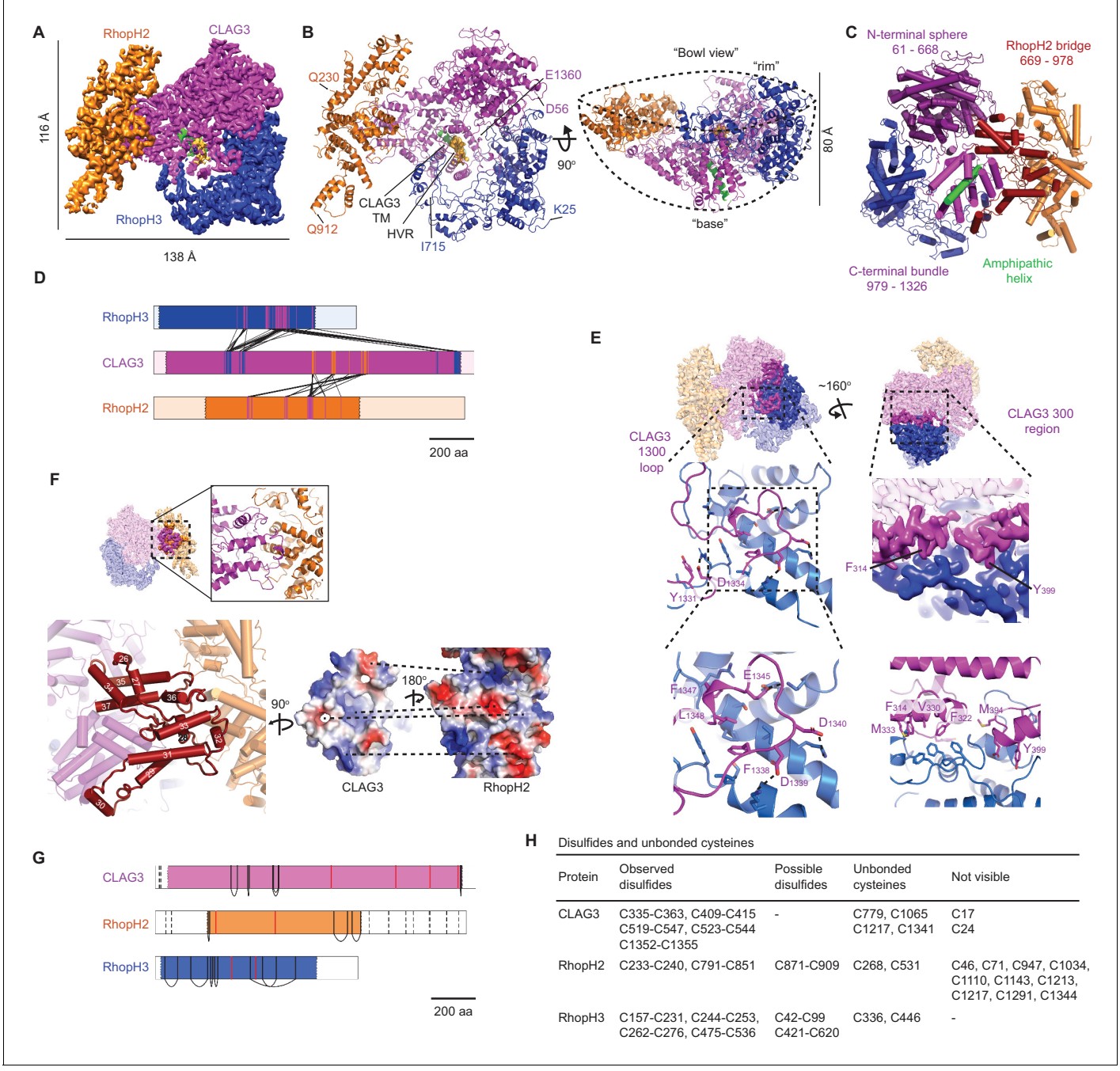

**Figure 2.** High-resolution structure of the soluble RhopH complex and stabilizing interactions. (A) Side view of the cryo-electron microscopy (cryo-EM) reconstruction with CLAG3, RhopH2, and RhopH3 color scheme maintained in all figures that show the structure. (B) Side and 90°-turned bowl ribbon diagrams of the RhopH complex. Buildable N- and C-terminal residues of each subunit are labeled. In (A) and (B), the CLAG3 HVR and single validated transmembrane α-helix are colored yellow and green, respectively. (C) CLAG3 domain architecture, with residues numbered from N-terminus. (D) Ribbon schematic illustrating pairwise interactions between subunits. The visualized N- and C-termini of each subunit are indicated by ribbon color change. (E) CLAG3–RhopH3 binding interface, as determined by the CLAG3 1300 loop and 300 regions, shown from separate angles. Enlarged views at bottom show critical CLAG3 residues involved in hydrophobic and charge–charge interactions. (F) The CLAG3–RhopH2 binding interface from different views. Enlarged image at bottom left shows CLAG3 α-helices that define the RhopH2 bridge, with helix numbering from one at the protein N-terminus. Right, Space-filling view of the CLAG3–RhopH2 surfaces at their binding interface. The proteins are separated from one another and rotated to expose the binding surfaces; blue and red shading reflect positive and negative electrostatic potential, respectively. Complementary surface potentials on these surfaces form salt bridges and contribute to tight interactions. (G) Ribbon schematic showing positions of cysteines that form intramolecular disulfides (black), unbonded cysteines (red), and cysteines that were not visualized (dashed black lines). Intermolecular disulfides were not observed. (H) Tabulated list of disulfides and unbonded cysteines.

*Figure 2 continued on next page*

*Figure 2 continued*

The online version of this article includes the following figure supplement(s) for figure 2:

**Figure supplement 1.** Cryo-electron microscopy (cryo-EM) data processing scheme.
**Figure supplement 2.** Observed and predicted movements of the RhopH complex.
**Figure supplement 3.** Multiple residues involved in CLAG3–RhopH2 binding.
**Figure supplement 4.** Conservation of amino acids at subunit interfaces.
**Figure supplement 5.** Neighbor-joining tree and structural similarities.

**Table 1.** Cryo-electron microscopy (cryo-EM) data collection, refinement, and validation statistics.

| | Freeze–thaw non-inserted (EMDB-22890) (PDB 7KIY) |
|---|---|
| **Data collection and processing** | |
| Microscope | Titan Krios |
| Camera | K2 Summit |
| Calibrated magnification | 59,500 |
| Voltage (kV) | 300 |
| Exposure time | 23.2 |
| Frame/total (s) | 2.5 frames/s |
| Number of frames per image | 58 |
| Electron exposure (e$^-$/Å$^2$) | 69.6 |
| Defocus range (μm) | 0.5–3.5 |
| Pixel size (Å) | 0.82 |
| Box size (pixels) | 400 |
| Symmetry imposed | C1 |
| Initial particle images (no.) | 311,390 |
| Final particle images (no.) | 68,216 |
| Map resolution (Å) | 2.92 |
| FSC threshold | 0.143 |
| Map resolution range (Å) | 2.89–12.10 |
| **Refinement** | |
| Initial model used (PDB code) | na |
| Model resolution (Å) | 3.06 |
| Model resolution range (Å) | 2.9–12.10 |
| Map sharpening B factor (Å$^2$) | −34 |
| Model composition | 19,943 |
| Nonhydrogen atoms | 2388 |
| Protein residues | 0 |
| Ligands | |
| B factors (Å$^2$) | 43.95 |
| Protein | Na |
| Ligand | |
| R.m.s. deviations | 0.012 |
| Bond lengths (Å) | 1.33 |
| Bond angles (°) | |
| Validation | 2.51 |
| MolProbity score | 13.52 |
| Clashscore | 1.42 |
| Poor rotamers (%) | |
| Ramachandran plot | 78.25 |
| Favored (%) | 19.38 |
| Allowed (%) | 2.36 |
| Disallowed (%) | |

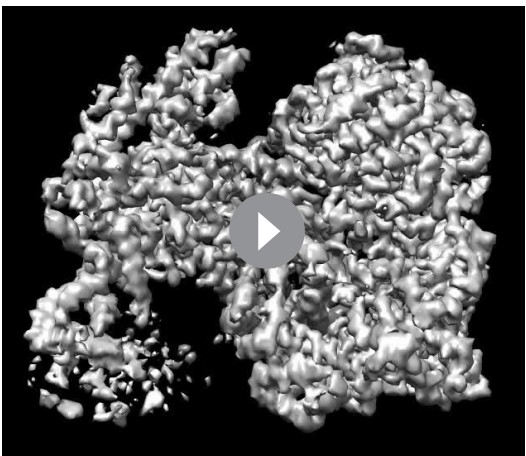

**Video 1.** First (inward) component of motion derived from multibody analysis.
https://elifesciences.org/articles/65282#video1

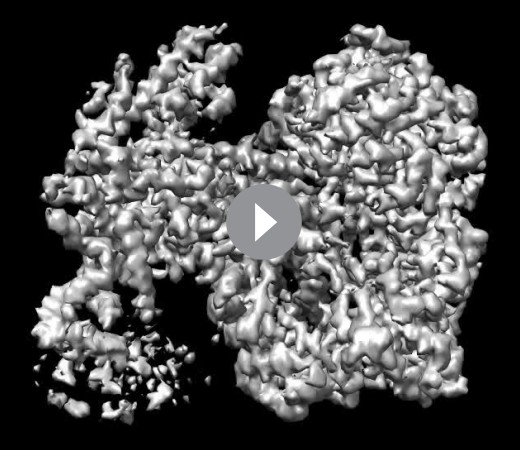

**Video 2.** Second (torsional) component of motion derived from multi-body analysis.
https://elifesciences.org/articles/65282#video2

## Subunit interactions and roles

CLAG3 contains three visually distinct domains (*Figure 2C*): an N-terminal sphere, an elongated central bridge for binding RhopH2, and a C-terminal bundle encasing an amphipathic helix that later integrates in the host erythrocyte membrane (*Sharma et al., 2015*). The N- and C-terminal domains hold RhopH3 tightly through bidentate interactions via a '1300 loop' and a '300 region' that form orthogonal pincer-grasp interactions. We illustrate these high-confidence interactions between CLAG3 and the other subunits in *Figure 2D*.

The 1300 loop packs against RhopH3 with discrete foci of hydrophobic and salt-bridge interactions (formed by CLAG3 residues Y1331, F1338, F1347, L1348 and D1334, D1339, D1340, E1345, respectively, *Figure 2E*, left panels). These CLAG3 residues and the cognate RhopH3 residues are highly conserved (*Figure 2—figure supplement 4B and D*), implicating essential roles in stabilizing the complex. The less strictly conserved 300 region consists of three α-helices, with two helices (10 and 11) interacting with RhopH3 residues 397–412 and 575–588 to create a hydrophobic core with a convergence of aromatic side chains (core formed by F314, F322, V330, M333, M394, Y399, *Figure 2E*, bottom right). The third CLAG3 helix (helix 14) and an upstream loop are closely apposed to RhopH3 through complex interactions. Together, the 300 region and 1300 loop produce an extensive 3700 Å$^2$ CLAG3 interface with RhopH3.

The 2005 Å$^2$ CLAG3–RhopH2 interface is much more fragmented (residues 706–715, 787–805, and 920–939 of CLAG3 and 414–435, 580–594, 682–708, and 760 from RhopH2). Interestingly, the CLAG3 backbone threads back and forth through the bridge domain (*Figure 2F*, bottom left) to form a wall-like interface; both surfaces are enriched with hydrophobic, conserved residues that form stable interactions (*Figure 2F*, bottom right; *Figure 2—figure supplements 3* and *4A,C*).

Each subunit has numerous conserved cysteines that contribute to tight assembly of this large complex through the formation of observed and possible disulfide bonds (*Figure 2G,H*). Although we did not detect

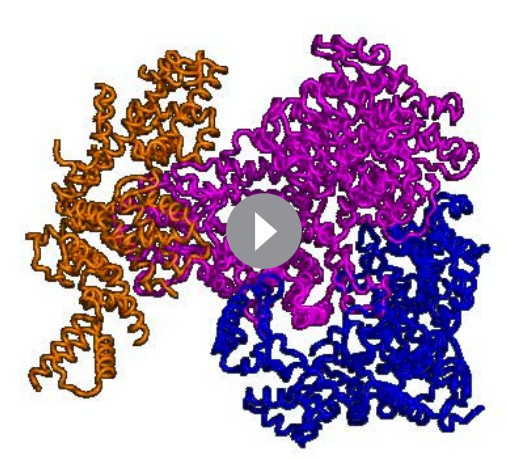

**Video 3.** First structural movement predicted by elNémo normal mode analysis.
https://elifesciences.org/articles/65282#video3

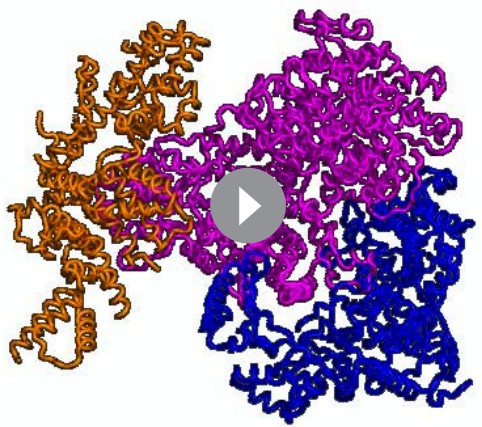 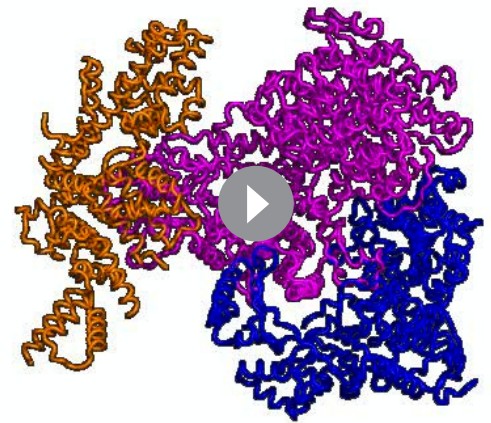

**Video 4.** Second structural movement predicted by elNémo normal mode analysis.

https://elifesciences.org/articles/65282#video4

**Video 5.** Third structural movement predicted by elNémo normal mode analysis.

https://elifesciences.org/articles/65282#video5

bonding between subunits, several cysteines were not visualized and could form such interactions. Conserved cysteines are a common feature of rhoptry proteins (*Kaneko, 2007*); they presumably contribute stability during egress and erythrocyte invasion and may also be critical for RhopH enzymatic activity at its final erythrocyte membrane destination (*Carter, 1973*).

CLAG3's central position in the structure, together with its surface exposure on erythrocytes and immune selection (*Iriko et al., 2008*), likely accounts for expansion of the *clag* gene family in all Plasmodium spp. We examined CLAG phylogeny and found that *P. falciparum* paralogs cluster into well-supported groups containing species infecting other mammals (*Figure 2—figure supplement 5A*). CLAG9 clustered independently and represented an older lineage. Sequences from Plasmodium spp.-infecting birds formed a separate group (labeled 'Clade F' in *Figure 2—figure supplement 5A*, based on taxonomy proposed by *Galen et al., 2018*). These sauropsid CLAG sequences are split into two well-supported orthologous groups, one that is basal to the CLAG2/CLAG3/CLAG8 orthologs and one that is basal to the CLAG9 orthologs. This pattern suggests an ancient split into

two paralogs in the common ancestor of sauropsid and mammalian Plasmodium spp., with subsequent diversification of mammalian paralogs. This diversification and ongoing gene family expansion (*Otto et al., 2018*) may yield distinct RhopH complexes capable of divergent functions including erythrocyte invasion, cytoadherence, and nutrient uptake. Expansion may also permit fine-tuning of PSAC permeabilities to allow nutrient uptake in both malnourished and well-fed hosts (*Mira-Martínez et al., 2017*).

Structural similarity searches of the Protein Data Bank (PDB) revealed weakly significant hits for each subunit that may guide structure–function studies of this Plasmodium-restricted complex (*Figure 2—figure supplement 5B and C*). RhopH3 exhibited the greatest structural similarity with alignment to domains from SepL, a regulator of type III translocon-based secretion in bacteria (*Burkinshaw et al., 2015*). RhopH2 partially aligned with Bcl-xL, an anti-apoptotic

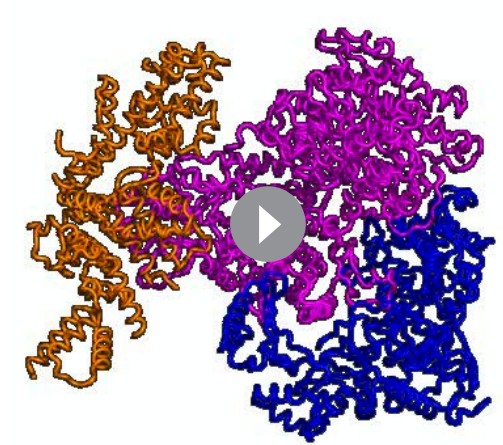

**Video 6.** Fourth structural movement predicted by elNémo normal mode analysis.

https://elifesciences.org/articles/65282#video6

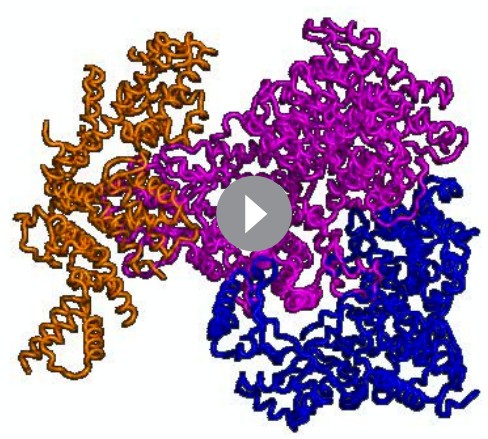

**Video 7.** Fifth structural movement predicted by elNémo normal mode analysis.
https://elifesciences.org/articles/65282#video7

protein that also regulates membrane permeabilization (*Finucane et al., 1999*). Both hits from our structural similarity searches raise the tantalizing possibility that RhopH2 and RhopH3 function to regulate PSAC. Such regulation could produce the unprecedented selectivity of this channel, which imports diverse nutrients including purines, amino acids, sugars, and some vitamins while maintaining very low $Na^+$ permeability to prevent host cell osmotic lysis (*Cohn et al., 2003*).

## Transmembrane domains are shielded in the soluble complex

Biochemical studies point to a direct contribution of the RhopH complex in PSAC-mediated nutrient uptake (*Gupta et al., 2018*; *Gupta et al., 2020*), with a single confidently predicted CLAG3 transmembrane domain distal to a 10–30 residue hypervariable region (HVR, *Figure 3A*). Site-directed mutagenesis of a conserved A1215 residue in this transmembrane domain (α-helix 44 in our structure) alters channel gating, selectivity, and conductance, supporting a pore-lining helix (*Sharma et al., 2015*). Notably, a PDB structure search identified this and several neighboring helices with a significant alignment to APH-1, an integral membrane component of human γ-secretase (*Figure 3—figure supplement 1A*). The corresponding APH-1 α-helix makes stable interactions with phospholipid in that structure (*Bai et al., 2015*), further supporting membrane insertion of CLAG3 α-helix 44.

This important helix is buried within a CLAG3 C-terminal bundle (*Figure 3B–D*), paralleling buried hydrophobic helices in some much smaller pore-forming proteins (*Dal Peraro and van der Goot, 2016*; *Figure 3—figure supplement 1B*). Transverse and longitudinal views establish that multiple Phe side chains segregate to one surface of helix 44 and that polar side chains line up at the opposite face (*Figure 3E*), as expected for a helix that lines an aqueous pore (*Sharma et al., 2015*). Although its physicochemical properties are conserved in CLAG orthologs, helix 44 exhibits little primary sequence conservation (*Figure 3F*). In contrast to this helix, the nearby HVR was poorly ordered, consistent with an unstructured extracellular loop that functions as an immune decoy (*Figure 3D*). The single predicted transmembrane domains on RhopH2 and RhopH3 are also buried in the soluble structure (helices defined by V740-D757 and G595-Y622 of these subunits, respectively; *Figure 3—figure supplement 1C–F*). Thus, known and predicted transmembrane domains are shielded in the trafficking RhopH complex (*Figure 3G*), implicating large-scale protein rearrangements for their membrane insertion.

## RhopH is synthesized as a non-integral complex

The peripheral and integral membrane pools (*Ito et al., 2017*) of the RhopH complex may both be formed during protein synthesis. Alternatively, the complex may be produced exclusively as a soluble form for trafficking and membrane insertion at a later point in the cell cycle. To distinguish between these models, we performed fractionation studies with synchronous cultures at defined developmental stages. During stage-specific synthesis in schizont-infected cells (*Ling et al., 2004*), both peripheral and integral membrane pools were reproducibly detected (*Figure 4A*, top row). This finding's interpretation is complicated by preexisting CLAG3 derived from the preceding cycle and trafficked to the infected cell surface (*Figure 1A*). To address this uncertainty, we treated early schizont-stage cultures with protease to identify prior-cycle CLAG3 inserted at the erythrocyte membrane. As the integral pool was quantitatively proteolyzed (*Figure 4A,B*), we conclude that the integral pool in these cells reflects protein made in the previous cycle; the larger carbonate-extractable pool represents newly synthesized protein.

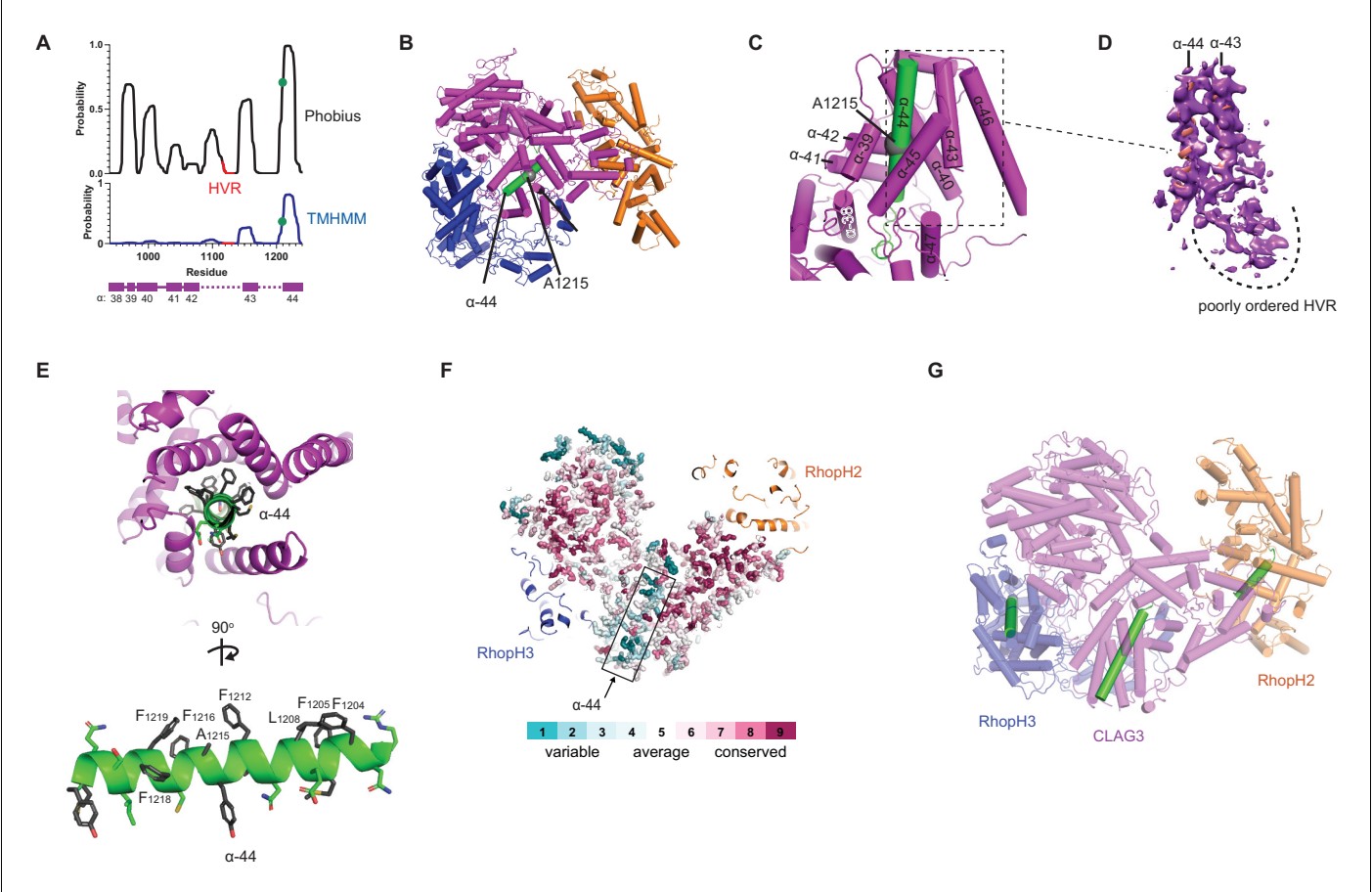

**Figure 3.** A CLAG3 transmembrane helix is buried in the soluble complex. (A) Posterior probability plots for transmembrane (TM) domain prediction, determined for residues 940–1240 of CLAG3 using indicated algorithms. Green circle on the plots' single confidently predicted TM (α-helix 44) represents A1215; HVR, hypervariable region. (B) Cylinder view of RhopH complex map showing the buried α-helix 44 (green). (C) Enlarged and turned view from (B). Additional helices that may interact with membranes are labeled. (D) Corresponding cryo-electron microscopy (cryo-EM) density and an adjacent poorly ordered HVR. (E) Top and side views of CLAG3 α-44. Note that hydrophobic side chains cluster on the upper helix surface in these views; polar residues are at the opposite surface and may line the eventual pore. (F) Slice-through view showing a thin interior section of the RhopH complex. CLAG3 is shown as sticks and colored by Consurf conservation score for each residue. Note that α-44 exhibits higher sequence variation than neighboring domains. (G) Cylinder view with known and predicted TM helices in green; these helices are buried and physically separated from one another.

The online version of this article includes the following figure supplement(s) for figure 3:

**Figure supplement 1.** Predicted transmembrane (TM) domains are buried in the soluble RhopH complex.

Fractionation studies using purified merozoites revealed carbonate-extractable CLAG3 and undetectable levels of integral protein (*Figure 4C*), consistent with packaging of newly synthesized RhopH complex into rhoptries and jettisoning of the prior-cycle integral host membrane pool upon schizont rupture; the host membrane marker, Band3, is also discarded at egress. Thus, CLAG3 is synthesized as a soluble protein that associates with other RhopH subunits to interact peripherally with membranes in rhoptries; whorls seen in rhoptries may provide a membranous surface for transfer of these proteins to the next erythrocyte (*Bannister et al., 1986*).

We then tracked this newly synthesized pool through the parasite bloodstream cycle and found that merozoites transfer their peripheral CLAG3 pool to immature ring-stage parasites, which also carry negligible amounts of the integral form (*Figure 4D,E*, rings). With parasite maturation, CLAG3 transitions from a primarily extractable form upon synthesis in schizonts into a growing integral pool after transfer into new erythrocytes (*Figure 4D,E*, trophozoites). During this conversion, CLAG3

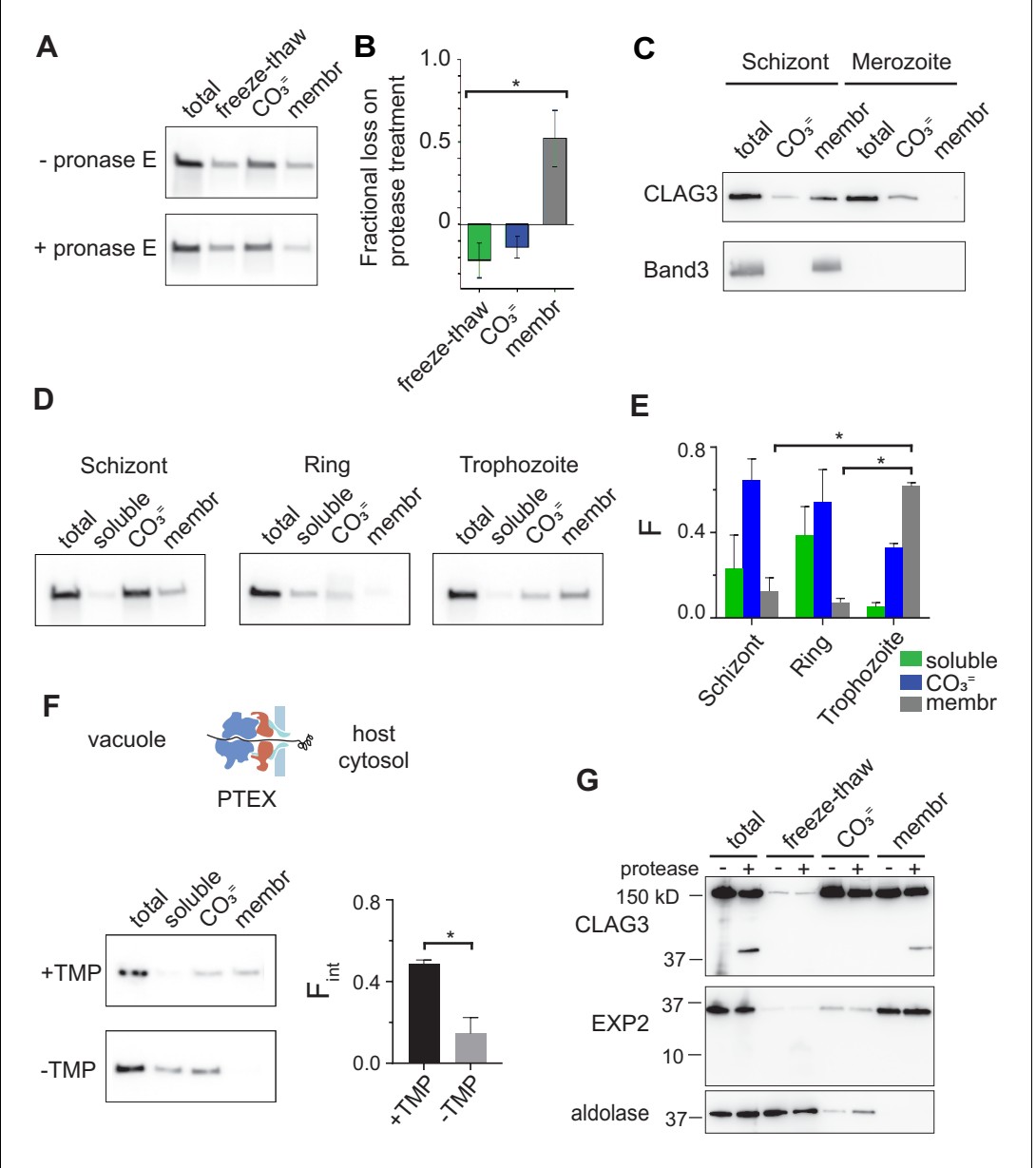

**Figure 4.** RhopH is produced as a soluble complex and requires interaction with the PTEX translocon for membrane insertion. (**A**) Immunoblot showing that pretreatment of mature schizont-infected cells with pronase E, a broad specificity protease, reduces CLAG3 in the membrane fraction (membr) without affecting freeze–thaw released or $Na_2CO_3$-extractable ($CO_3^=$) pools. (**B**) Mean ± S.E.M. fractional reduction of indicated CLAG3 pools upon pronase E treatment, determined from changes in band intensities from matched immunoblots as in (**A**). *p = 0.01, n = 3. (**C**) Immunoblots showing membrane fractionation of CLAG3 and Band3, a host membrane marker, in purified merozoites and their schizont-infected progenitor cells. Representative of two independent trials. (**D**) Similar fractionation studies at indicated stages throughout the *P. falciparum* bloodstream cycle. While schizont- and trophozoite-infected cells were enriched by the percoll–sorbitol method, ring-infected cells cannot be similarly enriched, presumably accounting for CLAG3 detection in the soluble lane and non-additive fractionation in rings. (**E**) Mean ± S.E.M. band intensities from three independent trials as in (**D**). *p < 0.005. (**F**) Schematic shows PTEX-mediated protein translocation and refolding in host erythrocyte cytosol. Middle, Anti-CLAG3 immunoblots from 13F10 cellular fractions with and without trimethoprim (TMP) (top and bottom blots, respectively). Bar graph shows mean ± S.E.M. fraction of integral membrane CLAG3 ($F_{int}$), determined from band intensities. *p < 0.015; n = 3. (**G**) CLAG3-tv2 fractionation studies using enriched mature infected cells. Top, Anti-HA blot showing that soluble CLAG3 (freeze–thaw and $CO_3^=$ lanes) is not susceptible to extracellular protease, but the integral pool (membr) is. The ~40 kDa cleavage product remains membrane embedded. EXP2, an intracellular

*Figure 4 continued on next page*

*Figure 4 continued*
parasite membrane protein, is primarily integral and is protease insensitive. Aldolase, a parasite cytosolic protein, is quantitatively released by freeze–thaw and carbonate treatment. Representative of more than three trials.

remains associated with other RhopH subunits and eventually localizes to the infected host cell membrane (*Vincensini et al., 2008*; *Nguitragool et al., 2011*; *Ahmad et al., 2020*).

How does this 440 kDa soluble RhopH complex convert into an integral form? Upon erythrocyte invasion, these and other rhoptry proteins are deposited into the parasitophorous vacuole. The PTEX protein translocon exports proteins secreted by the intracellular parasite into host cytosol (*de Koning-Ward et al., 2009*; *Beck et al., 2014*; *Ho et al., 2018*). It may therefore also export RhopH proteins into host cytosol; such transfer would be novel as it has not been established for other merozoite proteins deposited in the vacuole. While two studies have obtained conflicting results about whether RhopH proteins are exported via this translocon, both reported that PTEX knockdown abolishes activation of PSAC-mediated nutrient uptake at the host membrane (*Beck et al., 2014*; *Ito et al., 2017*). To examine membrane insertion, we performed CLAG3 fractionation using 13F10, a conditional PTEX knockdown parasite (*Beck et al., 2014*) whose protein export requires trimethoprim (TMP, *Figure 4F*). We found that CLAG3 transitions to an integral form in this parasite normally in the presence of TMP, but that PTEX knockdown produces a loss of integral CLAG3 (-TMP, p = 0.01, n = 3). CLAG3 that failed to insert into the membrane was more readily solubilized (-TMP, soluble lane), possibly due to protein crowding as a result of blocked export from the parasitophorous vacuole. Thus, RhopH membrane insertion is dependent on PTEX activity.

Stage-dependent membrane insertion was further evaluated in CLAG3-tv2 parasites with protease susceptibility studies. Both the freeze–thaw released and carbonate-extractable pools of CLAG3 were unaffected by extracellular protease, but the integral pool at the host membrane yielded a C-terminal cleavage product that remained membrane embedded (*Figure 4G*). α-Helix 44 is within this cleavage fragment and likely provides the responsible transmembrane anchor. Collectively, these findings indicate that CLAG3 is synthesized and trafficked in a soluble RhopH complex that undergoes marked rearrangements during its export to enable insertion at the host membrane.

## Discussion

We propose that RhopH evolved as a modular three-protein complex suited for essential and divergent functions at separate points in the bloodstream parasite cycle (*Figure 5*). A soluble form, packaged into rhoptry secretory organelles, facilitates RhopH3 contribution to erythrocyte invasion through still unknown mechanisms that presumably involve surface interactions. A large exposed surface area of ~32,000 Å$^2$ and globular architecture of RhopH3 provide candidates for inquiry. Our structure similarity searches found that RhopH3 residues 434–665 align with domains 2 and 3 of SepL; because domain three mediates interaction with the Tir receptor (*Burkinshaw et al., 2015*), one possibility is that RhopH3 interacts with an unidentified host cell receptor at this site. The RhopH3 C-terminus provides another surface for the presumed interactions, as suggested by site-directed mutagenesis of serine 804 and by studies with a monoclonal antibody against a 134 aa recombinant fragment (*Doury et al., 1994*; *Ekka et al., 2020*). This entire region (residues 716–897) is not resolved in our structure and appears to be flexible. Invasion-inhibiting antibodies that bind here may directly or indirectly prevent essential interactions with a cognate receptor. These findings and recent structural studies of the Rh5-CyRA-Ripr (*Wright et al., 2014*; *Wong et al., 2019*) should enable structure-guided therapies targeting erythrocyte invasion, an Achilles heel in the parasite's bloodstream cycle.

A soluble RhopH complex may also facilitate transfer to new erythrocytes for a second role in PSAC-mediated nutrient uptake (*Nguitragool et al., 2011*). We determined that the complex is transferred to the new host cell and deposited in the parasitophorous vacuole in a soluble form. The member subunits may then be exported into host cell cytosol via PTEX, as suggested by confocal immunofluorescence assays showing blocked export of each RhopH subunit in PTEX knockdown parasites (*Ito et al., 2017*). Forward and reverse coimmunoprecipitation experiments also suggest that the RhopH complex directly interacts with PTEX to enter host cell cytosol (*de Koning-Ward et al.,*

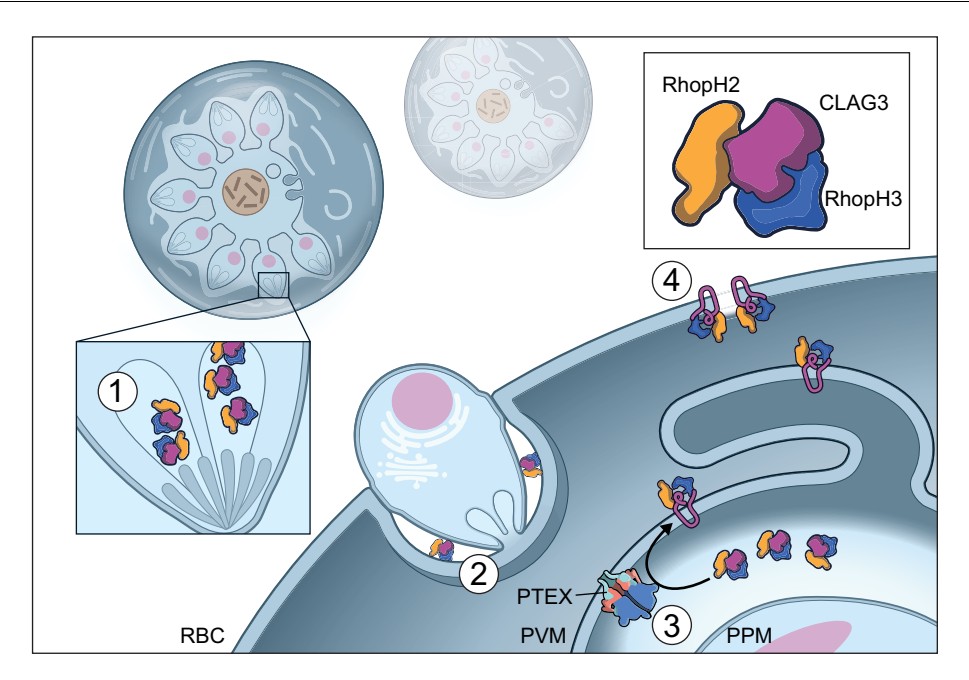

**Figure 5.** Model of RhopH synthesis and trafficking. The complex is produced in a soluble form and packaged into rhoptries (1) before transfer via extracellular merozoites to the nascent parasitophorous vacuole of a new host erythrocyte (2). The rhoptry may also contribute lipids to the nascent parasitophorous vacuole (*Dluzewski et al., 1995*). The soluble RhopH complex then crosses the PVM and undergoes membrane insertion via a PTEX-dependent mechanism (3). Finally, it is deposited on the host membrane with a small variant region on CLAG3 exposed to plasma, enabling channel-mediated nutrient uptake (4).

*2009*; *Counihan et al., 2017*). We next show that CLAG3 membrane insertion occurs via a PTEX-dependent mechanism (*Figure 4F*). Insertion may occur either concurrently with or after export. Because exported chaperones are thought to facilitate refolding of exported proteins and subsequent transit to specific host cell sites, failed CLAG3 membrane insertion may result from blocked export of multiple effector proteins.

Our structure reveals several intriguing and unique problems faced by the RhopH complex during its export and host membrane insertion. How this large complex crosses the parasitophorous vacuolar membrane remains unclear. If it transits directly through PTEX, this tightly assembled ternary complex with numerous disulfide bonds would require carefully coordinated unfolding and disassembly by HSP101 and possibly other vacuolar activities before translocation (*Beck et al., 2014*; *Ho et al., 2018*; *Matthews et al., 2019*). Subsequent reassembly in host cytosol may be even more complicated, with largely uncharacterized machinery needed to reform a stable complex without denaturation.

Another dilemma exposed by these studies is the precise mechanism by which one or more RhopH subunits become integral to the host erythrocyte membrane while remaining strictly associated with each other (*Ito et al., 2017*; *Ahmad et al., 2020*). Although membrane insertion during transit through PTEX would follow the precedent of Sec translocon-mediated membrane insertion in bacteria and other eukaryotes (*Denks et al., 2014*), PTEX appears to lack a lateral gate, as used by other translocons to transfer cargo proteins into the adjacent lipid bilayer (*Egea and Stroud, 2010*; *Corey et al., 2019*; *Ho et al., 2018*). We tend to favor membrane insertion after transfer into host cytosol. In this scenario, the energetically demanding process of conformational rearrangement to expose and insert specific α-helical domains into the host membrane may be facilitated by interactions with parasite-derived chaperones and Maurer's cleft organelles (*Proellocks et al., 2016*).

Although various studies support a role of the RhopH complex in PSAC formation and nutrient uptake (*Nguitragool et al., 2011*; *Mira-Martínez et al., 2019*; *Sharma et al., 2013*; *Ito et al., 2017*; *Counihan et al., 2017*), whether this ternary complex directly forms the aqueous pore in the

host erythrocyte membrane remains debated. In vitro selection previously implicated a short, but critical amphipathic CLAG3 motif in solute selectivity and PSAC single-channel gating (*Lisk et al., 2008*; *Sharma et al., 2015*). Our de novo structure establishes that this motif indeed forms an α-helix with hydrophobic and polar side chains segregated to opposite faces (helix 44, *Figure 3*), supporting a pore-lining helix in the host membrane. While studies suggest that CLAG3 oligomerizes at the host membrane and has an surface-exposed variant region (*Figure 5*; *Gupta et al., 2018*; *Nguitragool et al., 2014*), RhopH2 and RhopH3 are not exposed based on protease susceptibility studies (*Ito et al., 2017*).

The CLAG3 helix 44 and the individual predicted transmembrane domains on RhopH2 and RhopH3 are separated from one another by 46–101 Å in the soluble structure (*Figure 3G*). If all three helices come together to form the eventual nutrient pore, a remarkable rearrangement of the complex will be required during its conversion from a soluble to a membrane-inserted form. While our findings suggest interactions with PTEX or exported chaperone proteins, these rearrangements may also be facilitated by post-translational modifications such as site-specific phosphorylation and lysine acetylation (*Cobbold et al., 2016*; *Pease et al., 2013*).

Our findings provide a framework for understanding two unique and essential functions in bloodstream malaria parasites. Structure-guided development of therapies can now be pursued against a strictly conserved target exposed to plasma at two key points in the parasite cycle.

# Materials and methods

## Key resources table

| Reagent type (species) or resource | Designation | Source or reference | Identifiers | Additional information |
|---|---|---|---|---|
| Strain, strain background (*Plasmodium falciparum*) | KC5 | • 10.1128/mBio. 02293–17 | | Wt control |
| Cell line (*Plasmodium falciparum*) | CLAG3-tv2 | This paper | | C-terminal His$_{10}$-FLAG-thrombin-TEV-HA-twinstrept-BC2 tag |
| Cell line (*Plasmodium falciparum*) | CLAG3-GFP | This paper | | C-terminal His$_8$-mGFP-FLAG-Twin Strep tag |
| Cell line (*Plasmodium falciparum*) | CLAG3-tv1 | This paper | | C-terminal 3xFLAG-3xHA-His$_8$-Strept II tag |
| Cell line (*Plasmodium falciparum*) | CLAG3-tv1+RhopH2-mV | This paper | | CLAG3-tv1 with tandem RhopH2 C-terminal mVenus tag |
| Cell line (*Plasmodium falciparum*) | 13F10 | • 10.1038/ nature13574 | | TMP-dependent HSP101 conditional knockdown |
| Antibody | Anti-CLAG3 (mouse polyclonal) | • 10.7554/eLife. 23485 | | (1:1000) |
| Sequence-based reagent | CLAG3 sgRNA | This paper | For CRISPR editing | 5'-TAAAAACACTAATAAGACCA-3' |
| Recombinant DNA reagent | pUF1-Cas9 | • 10.1038/nbt.2925 | | Cas9 expression |
| Recombinant DNA reagent | pL6 | • 10.1038/nbt.2925 | | sgRNA expression and homology cassette |
| Recombinant DNA reagent | pL7- CLAG3-tv2 | This study | | Modification of pL6 for parasite transfection |
| Recombinant DNA reagent | pL7-CLAG3-GFP | This study | | Modification of pL6 for parasite transfection |
| Recombinant DNA reagent | pL7-CLAG3-tv1 | This study | | Modification of pL6 for parasite transfection |
| Chemical compound, drug | DSM1 | BEI Resources Repository | Cat# MRA-1161 | |

*Continued on next page*

*Continued*

| Reagent type (species) or resource | Designation | Source or reference | Identifiers | Additional information |
|---|---|---|---|---|
| Chemical compound, drug | WR99210 | David Jacobus | | |
| Commercial assay or kit | Anti-FLAG M2 affinity agarose resin | Sigma–Aldrich | Cat# A2220 | |
| Commercial assay or kit | 3xFLAG peptide | Sigma–Aldrich | Cat# F4799 | |
| Commercial assay or kit | Ni-NTA Agarose resin | Qiagen | Cat# 30210 | |
| Commercial assay or kit | Zeba microspin desalting columns, 40 kDa MWCO | Thermo Scientific | Cat# 87764 | |
| Other | Gold-coated quartz emitter | This study | | Native mass MS study |
| Commercial assay or kit | SYPRO Orange | Thermo Scientific | Cat# S6650 | Protein stability assay (1:5000) |
| Other | Carbon film grids | Electron Microscopy Sciences | Cat# CF200-Cu | |
| Other | Quantifoil Cu 300 mesh grids | Electron Microscopy Sciences | Cat# Q3310CR1.3 | |
| Other | 4–15% Mini-PROTEAN TGX gel | Bio-RAD | Cat# 4561086 | |
| Software, algorithm | Thermo Xcalibur Qual Browser | Thermo Scientific | versions 3.0.63 and 4.2.47 | |
| Software, algorithm | UniDec | • 10.1021/acs. analchem.5b00140; • 10.1007/s13361-018-1951-9 | versions 3.2 and 4.1 | http://unidec.chem.ox.ac.uk/ |
| Software, algorithm | m/z | Proteometrics LLC | | |
| Software, algorithm | EPU | ThermoFischer | | |
| Software, algorithm | Latitude | Gatan Inc | | |
| Software, algorithm | RELION 2.0; RELION 3.0 | • 10.1016/j.jsb.2012.09.006 | | https://www3.mrc-lmb.cam.ac.uk/relion |
| Software, algorithm | MotionCor2 | • 10.1038/nmeth.4193 | | https://emcore.ucsf.edu/ucsf-software |
| Software, algorithm | Gctf | • 10.1016/j.jsb.2015.11.003 | | https://www2.mrc-lmb.cam.ac.uk/research/locally-developed-software/zhang-software/ |
| Software, algorithm | UCSF Chimera | • 10.1002/jcc.20084 | | https://www.cgl.ucsf.edu/chimera/ |
| Software, algorithm | Coot | • 10.1107/S0907444910007493 | | https://www2.mrc-lmb.cam.ac.uk/personal/pemsley/coot/ |
| Software, algorithm | PHENIX assign_sequence | • 10.1107/S2059798319011471 | | https://www.phenix-online.org/ |
| Software, algorithm | PHENIX real space refine | • 10.1107/S2059798318006551 | | https://www.phenix-online.org/documentation/reference/real_space_refine.html |
| Software, algorithm | JPred | • 10.1093/nar/gkv332 | | http://www.compbio.dundee.ac.uk/jpred/ |
| Software, algorithm | elNémo server | • 10.1093/nar/gkh368 | | http://www.sciences.univ-nantes.fr/elnemo/ |
| Software, database | PlasmoDB | • 10.1093/nar/gkn814 | | https://plasmodb.org/plasmo/ |

*Continued*

| Reagent type (species) or resource | Designation | Source or reference | Identifiers | Additional information |
|---|---|---|---|---|
| Software, algorithm | MAFFT server | • 10.1093/bib/bbx108 | | https://mafft.cbrc.jp/alignment/server/ |
| Software, algorithm | MEGA X | • 10.1093/molbev/msy096;10.1093/molbev/msz312 | | https://www.megasoftware.net/ |
| Software, algorithm | ConSurf server | • 10.1093/nar/gkw408 | | https://consurf.tau.ac.il/ |
| Software, algorithm | NCBI Protein BLAST | • 10.1093/nar/25.17.3389 | | https://blast.ncbi.nlm.nih.gov/Blast.cgi |
| Software, algorithm | Clustal Omega | • 10.1093/nar/gkz268 | | https://www.ebi.ac.uk/Tools/msa/clustalo/ |
| Software, algorithm | Pymol | Schrödinger, LLC | | https://pymol.org/2/ |
| Software, algorithm | Dali server | • 10.1093/bioinformatics/btz536 | | http://ekhidna2.biocenter.helsinki.fi/dali/ |
| Software, algorithm | ImageJ | • 10.1186/s12859-017-1934-z | | https://imagej.nih.gov/ij/index.html |
| Software, algorithm | SigmaPlot 10.0 | Systat | | |
| Software, algorithm | Prism 8.2 | GraphPad | | |

## Parasite culture

*P. falciparum* laboratory strains were grown in $O^+$ human erythrocytes (Interstate Blood Bank) using standard methods and maintained at 5% hematocrit under 5% $O_2$, 5% $CO_2$, 90% $N_2$ at 37°C.

## Endogenous tagging

CRISPR-Cas9 gene editing was used to produce engineered *P. falciparum* lines using the KC5 laboratory clone carrying a single *clag3h* gene to avoid epigenetic switching (Gupta et al., 2018). Transfections were performed by electroporation of pUF1-Cas9 and modified pL6 plasmids for homologous replacement of the genomic site as described (Ghorbal et al., 2014); 1.5 µM DSM1 and 2 nM WR99210 were used to select for integrants, which were detected by PCR. All experiments were performed with limiting dilution clones that were confirmed with DNA sequencing.

Primary protein purifications used the edited CLAG3-tv2 clone, in which a C-terminal multiple affinity tag consisting of $His_{10}$-FLAG-thrombin-TEV-HA-twinstrep-BC2 nanobody binding site was appended to an otherwise unmodified CLAG3h. The CLAG3-GFP incorporates a C-terminal $His_8$-monomeric GFP-FLAG-Twin strep tag on CLAG3h. The CLAG3-tv1+RhopH2-mV strain contains a C-terminal 3xFLAG-3xHA-$His_8$-Strept II tag on CLAG3 and a monomeric Venus tag at the RhopH2 C-terminus; this parasite was produced by sequential CRISPR-Cas9 editing of the two genomic loci and used for negative stain imaging of mVenus-tagged RhopH2.

## Protein purification

Up to 1 mL of enriched schizont-stage parasites were harvested by the percoll–sorbitol method and frozen in liquid nitrogen at 20% v/v in 200 mM NaCl, 10 mM Tris, pH 7.5 with 1 mM phenylmethylsulfonyl fluoride (PMSF). Frozen parasites were thawed at room temperature, and insoluble debris was pelleted at 20,000 × g for 10 min at 4°C. NaCl was added to 500 mM before overnight incubation of the clarified lysate with anti-FLAG M2 affinity agarose resin (Sigma–Aldrich) at 4°C with gentle agitation. The resin was subsequently washed with 1–5 mL of 10 mM Tris, pH 7.5 and 500 mM NaCl before elution in 10 mM Tris, pH 7.5, 200 mM NaCl and 0.15 mg/mL 3xFLAG peptide. The eluate was concentrated for native mass spectrometry and cryo-EM studies via a second affinity purification on Ni-NTA agarose resin (Qiagen) and small volume elution in 200 mM NaCl, 300 mM imidazole, 10 mM Tris, pH 7.5. After overnight dialysis to remove imidazole, purified RhopH complex was further concentrated by ultracentrifugation at 150,000 × g for 1 hr, yielding 0.8–2 mg/mL protein in 30 µL.

## Native mass spectrometry analysis

Purified RhopH complex was buffer-exchanged into native mass spectrometry (MS) solution (200 mM ammonium acetate, pH 7.5, 0.01% Tween-20) using Zeba microspin desalting columns with a 40 kDa cut-off (ThermoScientific; *Olinares et al., 2016*; *Olinares and Chait, 2020*). Buffer-exchanged sample (3 µL) was loaded into a locally prepared gold-coated quartz emitter and electrosprayed into an Exactive Plus EMR instrument (ThermoFisher Scientific) with a modified static nanospray source (*Olinares and Chait, 2020*). The MS parameters used include spray voltage, 1.2–1.3 kV; capillary temperature, 150–250℃; in-source dissociation, 10 V; S-lens RF level, 200; resolving power, 17,500 at m/z of 200; AGC target, $1 \times 10^6$; maximum injection time, 200 ms; number of microscans, 5; injection flatapole, 8 V; interflatapole, 4 V; bent flatapole, 4 V; high-energy collision dissociation, 200 V; ultrahigh vacuum pressure, $7–8 \times 10^{-10}$ mbar; total number of scans, $\geq$100. Mass calibration in positive extended mass range (EMR) mode was performed using cesium iodide.

The acquired MS spectra were visualized using Thermo Xcalibur Qual Browser (versions 3.0.63 and 4.2.47). Spectra deconvolution was performed either manually or using the software UniDec versions 3.2 and 4.1 (*Marty et al., 2015*; *Reid et al., 2019*). The resulting deconvolved spectrum from UniDec was plotted using the m/z software (Proteometrics LLC). Experimental masses were reported as the mean ± SD across all calculated mass values within the observed charge state series. Mass accuracies were calculated as the % difference between the measured and expected masses relative to the expected mass.

## Protein thermostability

Thermal denaturation of the RhopH complex was evaluated with two methods. ThermoFluor assays were performed with 20 µL of 0.4 mg/mL freeze–thaw extracted RhopH complex and a 1× dilution of SYPRO Orange. Fluorescence intensity was continuously monitored during a thermal ramp from 25℃ to 95℃ in 0.5℃/10 s increments. Raw fluorescence and first-derivative plots were used to assess unfolding. RhopH complex aggregation was also evaluated using sizing with thermal ramp application on Uncle (Unchained Labs) and duplicate samples of 8.9 µL of 0.1 mg/mL RhopH complex. Aggregation was measured by monitoring static light scattering at 266 and 473 nm with a ramp from 20℃ to 80℃ at a constant rate of 1.0℃/min for 1 hr with measurements at 0.5℃ increments.

## Negative stain data acquisition

Purified RhopH protein (4.8 µL of a 0.05 mg/mL solution) was applied to carbon film grids (CF200-Cu, Electron Microscopy) and stained with 4.8 µL of 0.75% uranyl formate for 30 s. After drying, grids were loaded onto a ThermoFischer Tecnai 12 electron microscope with a Gatan Ultra Scan camera operating at 120 kV. Images were collected using EPU software (ThermoFischer) at 67,000× magnification for a pixel size of 1.77 Å. The datasets consisted of between 69 and 142 micrographs (culture-media RhopH, 69 micrographs; complexes containing RhopH2-mV, 109; CLAG3-tv1, 124; CLAG3-GFP, 142).

## Negative stain image processing

All negative stain image processing was performed using RELION 2.0 (*Scheres, 2012*). Micrographs were processed without CTF correction. Initial auto-picking was performed using a Gaussian blob. Well-behaved classes from 2D classification of Gaussian blob-picked particles were used for template-based auto-picking. Further 2D classification was performed to clean the particle set. For datasets with GFP derivative tagging, additional density for the bulky epitope was visible is several 2D classes. For freeze–thawed solubilized and spend-media RhopH, an initial model was generated and used for 3D auto-refinement in RELION. Three-dimensional models represent views in Chimera (*Pettersen et al., 2004*).

## Cryo-EM data acquisition

2.5 µL of 0.8 mg/mL RhopH was applied to glow-discharged Quantifoil Cu 300 mesh grids (1.2/1.3), blotted for 3 s, and plunge frozen in liquid ethane cooled by liquid nitrogen using a Vitrobot plunge freezing instrument (FEI/ThermoFisher). The blotting chamber was maintained at 20℃ and 100% humidity. One thousand three hundred and ten micrographs were collected on a Titan Krios

(ThermoFisher) transmission electron microscope operated at 300 kV. Images were recorded on a K2 Summit camera (Gatan Inc) operated in super-resolution counting mode and a physical pixel size of 0.84 Å. The detector was placed at the end of a GIF Quantum energy filter (Gatan Inc), operated in zero-energy-loss mode with a slit width of 20 eV. Each image was fractionated into 58 frames with a frame exposure of 0.4 s and a dose rate of 3 $e^-/Å^2$/s, giving a total accumulated dose of 70 $e^-/Å^2$ over the 23.2 s exposure. All data was collected using the Latitude S software (Gatan Inc).

## Cryo-EM image processing

All cryo-EM image processing was performed in RELION 3.0. Movies were motion corrected and dose-weighted using MotionCor2 (*Zheng et al., 2017*). Contrast transfer function (CTF) parameters were determined using the Gctf (*Zhang, 2016*) wrapper in RELION. Initial particle picking was performed with the Laplacian-of-Gaussian (LoG) picker in RELION. Subsequent 2D classes from the LoG-picked particles were used for template-based auto-picking performed in RELION resulting in 311,390 particles. After two rounds of 2D classification, the initial collection was cleaned to 214,233 particles and used to generate an initial 3D model. Three-dimensional classification using five classes with regularization parameter T = 4 resulted in one well-resolved class of 68,216 particles. Three-dimensional auto-refinement of these particles resulted in a 3.26 Å map. Two rounds of particle polishing and one round of CTF refinement further improved the resolution to 2.92 Å. Although the large lobe was well-resolved and permitted de novo model building, the small lobe and C-terminal bundle of CLAG3 were resolved to lower resolution inhibiting interpretation. Further 3D classification did not improve small subunit interpretability. To better resolve RhopH2 and CLAG3 C-terminal domain, multibody refinement was performed (*Nakane et al., 2018*). Multibody refinement using masked region 1 of the large subunit and masked region two as the small subunit and the bridge between the large and small subunit resulted in better EM density for mobile elements of the small subunit although a lower overall resolution for the second masked region. Multibody analysis also yielded the top components of motion.

## Model building and refinement

Model building was performed in Coot (*Emsley et al., 2010*). EM density maps were generated in RELION by post-processing with a constant B factor or locally sharpened regions of the maps in Local Resolution. Initially, a poly-alanine model was built for well-ordered regions of the RhopH complex in Coot. The sequence registry was determined by a combination of manual examination of side-chain density and the PHENIX assign_sequence program (*Liebschner et al., 2019*), which predicts sequence registry based on side-chain density. Regions of the map with low resolution were built through a combination EM density interpretation and secondary structure prediction performed in JPred (*Drozdetskiy et al., 2015*). Real space refinement with secondary structure restraints was performed in PHENIX real space refine (*Afonine et al., 2018*). Structural figures were generated in PyMOL 2.1.0 (Schrödinger) or Chimera. Prediction of motion in the final model was performed using the elNémo server (*Suhre and Sanejouand, 2004*).

## Phylogenetic analysis

CLAG DNA sequences were downloaded from PlasmoDB (http://PlasmoDB.org) and aligned using the MAFFT server (*Katoh et al., 2019*) with default parameters. Sequences shorter than 2000 nucleotides in length were removed to maximize sequence overlap. The multiple-sequence alignment was corrected manually to preserve the reading frame. Phylogenetic analysis of the remaining 147 sequences was performed using the MEGA X software (*Kumar et al., 2018*; *Stecher et al., 2020*). A phylogenetic tree was inferred using the neighbor-joining method (*Saitou and Nei, 1987*) based on pairwise distances computed using the maximum composite likelihood method (*Tamura et al., 2004*), with the rate variation among sites modeled with a gamma distribution (shape parameter = 1). To assess how well the data supported the groups in the tree, 250 bootstrap replicates were performed (*Felsenstein, 1985*).

## Conservation analysis

The ConSurf server (https://consurf.tau.ac.il/; *Ashkenazy et al., 2016*) was used to generate per-residue conservation scores and map conservation values on the 3D RhopH complex structure. Non-

redundant sequences of RhopH subunits from Plasmodium spp. were identified through the use of PlasmoDB and NCBI Protein BLAST. The sequences were aligned using Clustal Omega. ConSurf was then used to evaluate evolutionary conservation of amino acid residues; the resulting conservation scores were used for color-coding residues in PyMOL.

### Structural similarity searches

The Dali server (http://ekhidna2.biocenter.helsinki.fi/dali/; *Holm, 2019*) was used to search for proteins with 3D structures like that of the RhopH complex. Exhaustive PDB database searches revealed significant matches to specific domains from individual RhopH subunits, as defined by Dali Z-scores $\geq$ 3.0. PyMOL alignments of RhopH domains and PDB structures of corresponding hits were used to evaluate biological significance.

### Membrane fractionation

Synchronization for stage-dependent membrane fractionation assays utilized two 5% sorbitol treatments ~6 hr apart. Ring-stage infected cells were harvested immediately without enrichment. Trophozoite- and schizont-stage-infected cells were then harvested 18 hr and 40 hr after sorbitol treatment, respectively, and enriched through the percoll–sorbitol method. Cells infected with 13F10 growth with or without TMP were harvested without enrichment as these cells lack PSAC activity (*Beck et al., 2014*; *Ito et al., 2017*).

Freed merozoite studies were performed with 3D7 parasites using synchronous schizonts enriched using the percoll–sorbitol method. Purified schizonts were cultured with 25 µM E64D at 7.5 $\times$ 10$^7$ cells/mL and closely monitored for 4–5 hr for the development of segmenters containing fully formed merozoites. Cells were then washed, adjusted to 2.5 $\times$ 10$^7$ cells/mL in complete media, and allowed to recover at 37°C for 15 min. Freed merozoites (2.5 $\times$ 10$^8$ cells/mL) were obtained by sequential passage through two 1.2 µm syringe filters to rupture the mature segmenters. A hemocytometer was used to confirm that merozoites were free of contaminating intact erythrocytes before pelleting (4500 $\times$ g, 5 min) and freezing along with matched intact schizonts.

Fractionation studies were performed using matched cell pellets resuspended in lysis buffer (7.5 mM Na$_2$HPO$_4$, 1 mM EDTA, pH 7.5) at 3.5% hematocrit; this cell lysate corresponded to the 'total' input. Cellular debris and membranes were pelleted by ultracentrifugation at 100,000 $\times$ g for 1 hr at 4°C. The supernatant was kept as the 'soluble' fraction. Membranes were resuspended and incubated in 200 µL of 100 mM Na$_2$CO$_3$, pH 11 at 4°C for 30 min before ultracentrifugation (100,000 $\times$ g, 1 hr, 4°C) to separate peripheral from integral membrane proteins. Samples were neutralized with 1 M HCl and solubilized in a modified Laemmli buffer with a final 6% sodium dodecyl sulfate (SDS) concentration.

Protease susceptibility experiments used percoll–sorbitol-enriched cells. Infected cells were treated with Pronase E in phosphate-buffered saline (PBS) supplemented with 0.6 mM CaCl$_2$ and 1 mM MgCl$_2$ for up to 1 hr at 37°C. They were then extensively washed in PBS with 1 mM PMSF prior to membrane fractionation.

### Immunoblotting

Samples were prepared in a modified Laemmli buffer with a final 6% SDS concentration. Proteins were separated on a 4–15% Mini-PROTEAN TGX gel (Bio-RAD) and transferred to nitrocellulose. After blocking, antibodies against CLAG3 (*Nguitragool et al., 2011*), Band3 (Santa Cruz), HA epitope tag (Sigma–Aldrich), EXP2 (European Malaria Reagent Repository), or aldolase (Abcam) were applied and visualized as described (*Ito et al., 2017*). Band intensities were quantified using ImageJ and analyzed in Prism (GraphPad).

### Statistical analysis

Statistical significance for numerical data was calculated by unpaired Student's t-test or one-way ANOVA. Significance was accepted at p < 0.05 or indicated values.

## Acknowledgements

We thank Arasu Balasubramaniam for help with ThermoFluor assays, Anthony Armstrong for guidance on elNémo normal mode analysis, Ryan Kissinger and Anita Mora for artwork, Daniel Goldberg for the 13F10 clone, and David Jacobus for WR99210. DSM1 (MRA-1161) was obtained through MR4 as part of the BEI Resources Repository, NIAID, NIH. This work utilized the computational resources of the NIH HPC Biowulf cluster (http://hpc.nih.gov).

## Additional information

### Competing interests

Sriram Subramaniam: Reviewing editor, *eLife*. The other authors declare that no competing interests exist.

### Funding

| Funder | Grant reference number | Author |
|---|---|---|
| National Institute of Allergy and Infectious Diseases | | Sanjay A Desai |
| National Cancer Institute | | Sriram Subramaniam |
| National Institutes of Health | P41 GM103314 | Brian T Chait |
| National Institutes of Health | P41 GM109824 | Michael P Rout<br>Brian T Chait |
| Canada Excellence Research Chairs, Government of Canada | | Sriram Subramaniam |

The funders had no role in study design, data collection and interpretation, or the decision to submit the work for publication.

### Author contributions

Marc A Schureck, Conceptualization, Formal analysis, Investigation, Writing - original draft, Writing - review and editing; Joseph E Darling, Alan Merk, Jinfeng Shao, Paul Dominic B Olinares, Kurt Wollenberg, Formal analysis, Investigation, Writing - review and editing; Geervani Daggupati, Investigation, Writing - review and editing; Prakash Srinivasan, Supervision, Methodology, Writing - review and editing; Michael P Rout, Brian T Chait, Formal analysis, Supervision, Writing - review and editing; Sriram Subramaniam, Conceptualization, Resources, Formal analysis, Supervision, Writing - review and editing; Sanjay A Desai, Conceptualization, Formal analysis, Supervision, Writing - original draft, Project administration, Writing - review and editing

### Author ORCIDs

Paul Dominic B Olinares https://orcid.org/0000-0002-3429-6618
Sriram Subramaniam https://orcid.org/0000-0003-4231-4115
Sanjay A Desai https://orcid.org/0000-0003-2150-2483

### Decision letter and Author response

Decision letter https://doi.org/10.7554/eLife.65282.sa1
Author response https://doi.org/10.7554/eLife.65282.sa2

## Additional files

### Supplementary files

• Transparent reporting form

## Data availability

All data generated or analysed during this study are included in the manuscript and supporting files. Cryo-EM maps have been deposited in EMDB and PDB.

The following datasets were generated:

| Author(s) | Year | Dataset title | Dataset URL | Database and Identifier |
|---|---|---|---|---|
| Schureck MA, Darling JE, Merk A, Subramaniam S, Desai SA | 2021 | Plasmodium falciparum RhopH complex in soluble form | https://www.rcsb.org/structure/7KIY | RCSB Protein Data Bank, 7KIY |
| Schureck MA, Darling JE, Merk A, Subramaniam S, Desai SA | 2021 | Plasmodium falciparum RhopH complex in soluble form | https://www.emdatare-source.org/EMD-22890 | EMDataResource, EMD-22890 |

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
