## [Decision Letter]

**Acceptance summary:**

This study provides novel insights into the structure and trafficking of an essential protein complex of the malaria parasite. *Plasmodium falciparum* relies on the RhopH protein complex for invasion of erythrocytes and for nutrient uptake. Here, Schureck et al. use cryo-electron microscopy to determine the first 3D structure of the RhopH complex purified from the parasite. They provide compelling evidence that the complex is synthesized as a soluble form that is exported to the infected host cell and subsequently inserted into the erythrocyte membrane.

**Decision letter after peer review:**

[Editors’ note: the authors submitted for reconsideration following the decision after peer review. What follows is the decision letter after the first round of review.]

Thank you for submitting your work entitled "Malaria parasites use a soluble RhopH complex for erythrocyte invasion and an integral form for nutrient uptake" for consideration by *eLife*. Your article has been reviewed by four peer reviewers, one of whom is a member of our Board of Reviewing Editors, and the evaluation has been overseen by a Senior Editor. The reviewers have opted to remain anonymous.

Our decision has been reached after consultation between the reviewers. Based on these discussions and the individual reviews below, we regret to inform you that in its current form, your work will not be considered further for publication in *eLife*.

As you will see, the four reviewers are broadly supportive of this work, which addresses an important topic and provides the first atomic resolution cryo-EM structure of endogenous *P. falciparum* RhopH complex. However, they were especially concerned that the conclusions drawn from the structural and biochemical analysis are not sufficiently supported by the data. Importantly, they raised three main issues that will take some time to address (probably more than two months) and in consequence, we have together opted for a rejection with a strong encouragement for re-submission. If you are able to address these issues we will consider a newly submitted form of this paper that we will treat as a revised manuscript. If you choose this course of action please submit a separate cover letter detailing the changes you have made.

Main points to be addressed:

1) Issues regarding the structure quality should be addressed (Reviewer #3), and some clarification and precision in the structure description is required (Reviewers #2 and #3).

2) The four reviewers had concerns about the role of PTEX translocon in RhopH export. You should either provide direct evidence that RhopH interacts with PTEX, or revise your model.

3) A more thorough biochemical characterization of RhopH complex is required to strengthen your model, including experiments with pronase (Reviewers #2 and #4) or free merozoites (Reviewer #1).

Reviewer #1:

This is an interesting study of *Plasmodium falciparum* RhopH, a complex of three proteins (CLAG, RhopH2 and RhopH3) that plays a dual role in the malaria parasite, first in merozoites for invasion of erythrocytes, and then at the erythrocyte membrane for nutrient uptake by the parasite. The authors developed a freeze-thaw procedure for gentle detergent-free harvest of a soluble form of the RhopH complex, and used cryo-EM to determine de novo the structure of the complex at high resolution. Interestingly, predicted transmembrane domains are shielded in the soluble complex, suggesting a model where the complex is transferred to the erythrocyte as a soluble form, and then undergoes large-scale conformational changes for insertion in the host erythrocyte membrane. The authors then conducted a biochemical characterization of RhopH and report that the complex is synthesized as a non-integral complex, and that insertion in the erythrocyte membrane requires export through the parasite PTEX protein translocon.

While the structure data do not provide clues on the function of RhopH3 during invasion or on the structure of the integral membrane complex, this study provides a framework for future functional studies, and gives novel and important insights into how the malaria parasite exports a complex of rhoptry proteins into the host cell.

I do not have sufficient expertise to evaluate the quality of the cryo-EM work.

1) Cryo-EM as used to determine the structure of a soluble form of RhopH, which apparently corresponds to a minor fraction of the complex "not associated with membranes". It is not clear what exactly this fraction corresponds to. In a previous report (Ito et al., 2017) the same group indicated that members of the complex are membrane-associated, but extractable in part by carbonate. Why is this fraction not recovered after hypotonic lysis?

It is not clear whether the soluble form corresponds to peripheral membrane proteins.

Have the authors attempted recovering the complex from purified merozoites or spent media for cryoEM? The authors propose that a change in conformation explains the differential behavior of the complex, but they should discuss alternative explanations, such as post-translational modifications or protein interactions, which could alter the association of the complex with the membranes.

2) The biochemical characterization of the different pools of RhopH complex in schizonts is complicated by the coexistence of previously synthesized RhopH (integral form) and newly synthesized protein (peripheral, in the rhoptries). In Figure 4, the authors should include purified merozoites, which should contain only the peripheral/soluble complex, and erythrocyte ghost membranes, which should only contain the integral form. This would provide clearer evidence to support their model.

3) In the PTEX mutant experiment, it is crucial to analyze parasites at the same stage. Beck et al., 2014, showed that in the absence of TMP, 13F10 mutant parasites are blocked at the ring stage, so it is not surprising that the protein extraction profile in Figure 4E is similar to the ring profile in Figure 4C, and does not prove that export through PTEX is required for RhopH membrane insertion. This is a major weakness in the model.

Reviewer #2:

Schureck et al. present some interesting work on the structure of the soluble RhopH complex in *Plasmodium falciparum*. They use cryo-electron microscopy to show that clag3, RhopH2 and RhopH3 are present in a 1:1:1 stoichiometry, and that known and predicted transmembrane helices are shielded in the soluble complex. This work presents a significant advance in the field and in my opinion warrants publication in *eLife*.

Reviewer #3:

I see real merit in this study as it describes and reports the biochemical characterization and first atomic resolution cryo-EM structure of the ternary RhopH complex from the malaria parasite purified directly from endogenous source, using a strategy similar to what has been described by Ho and Beck for the PTEX vacuolar translocon. The manuscript however needs clarification and editing to make it more easily readable to a broader public in *eLife*.

Authors show that a soluble and remarkably stable 1:1:1 RhopH endogenous complex can be isolated directly from infected red blood cells using CRISPR-Cas9 engineered methods. Stoichiometry is determined by native MS combined with the NS-EM analysis of complexes where one subunit is selectively tagged with a “bulky” (relatively speaking compared to each subunit) GFP tag. The NS-EM and biochemical/MS-proteomics data support their claims and are clearly described and illustrated (Figure 1 and its supplementary figures).

An atomic resolution (~2.9Å) cryo-EM structure is described with a particular emphasis on the protein-protein interfaces and their conservation. Based on the nature, conservation and intricacy levels of these CLAG3-RhopH1 and CLAG3-RhopH2 surfaces the authors conclude that the soluble RhopH complex is robust and built to promote invasion and undergo trafficking throughout the cell to reach its final destination the red blood cell membrane.

3 TMDs are predicted, one in each subunit, all buried in this soluble form of the RhopH complex, a pro-PSAC form so to speak. They propose a model where this soluble RhopH converts into a membrane-integrated form in the RBCM with TMDs becoming inserted in the RBCM to form the functional PSAC transporting diverse solutes/nutrients.

Description and illustration of the structure is lacking clarity and precision in some key aspects (TMDs). The main weakness of the present work might reside in the proposed interaction between PTEX and RhopH for the trafficking of the latter throughout the PVM into the infected erythrocyte.

Abstract. Although the Abstract is clear it should introduce the names of the two other rhoptry proteins RhopH2 and RhopH3 proteins simultaneously within that third sentence for the sake of clarity and consistency, as it sounds odd to bring all the attention on CLAG3 (or RhopH1) but not mention the other two subunits so intimately associated with CLAG3.

Second paragraph in Introduction. I would clarify “expansion in humans and vertebrates” as in “*Plasmodium* spp. infecting humans and other vertebrates such as birds, rodents and primates” as they explain more clearly later in the manuscript.

At the end of the Introduction (and maybe also in the Abstract) I would suggest being a bit more explicit in introducing PTEX (I am aware that referencing the initial study from DeKoning-Ward et al. is done much later in the text) and its function as a vacuolar translocon. In general the Introduction assumes that most readers would be familiar with the exquisitely complicated life cycle of apicomplexan parasites such as *Plasmodium* (or Toxoplasma): Introducing the existence and formation of a host-derived parasitophorous vacuole (membrane) following invasion of the host red blood cell by the parasite sounds necessary to clarify the general picture. Again this is done quite late in the body of the manuscript.

1) Structure determination

End of Introduction. Results section and in general. Concerning the structure itself presented at 2.9Å resolution as per cryo-EM standards. 2.9 Å is not high resolution: atomic resolution should suffice.

Table 1. Statistics for cryo-EM structural analysis.

In Data collection and processing. Any reason why defocus, FSC threshold and map resolution values are not indicated?

In Refinement. Any reason why model resolution range is not indicated?

Going through the PDB preliminary validation report indicates the presence of a fair amount of clashes in this large structure. RhopH2 in particular seems less well defined.

Upon inspection of the structure using the provided PDB file, I will comment about disulfide bonds and "free" cysteines (reduced). The three proteins contain a fair amount of cysteines and including a total of 7 clearly identified disulfide bonds (overall the 3 subunits) as one could expect in "secreted" proteins facing harsher conditions and as the Authors say "suited for trafficking and transfer to a new host erythrocyte". These are fairly complex patterns of disulfide linkages that did not seem to catch the attention of the Authors.

In CLAG3 (chain A in the PDB file) they are 14 cysteines.

with lone cysteines C779 C1065 C1217 C1431

and three listed disulfide bonds (C409-C415) (C523-C544) and (C1352-C1355)

but pairs C335-?-C363 and C519-?-C547 seem also close enough to form disulfide bonds.

In RhopH2 (chain B in the PDB file) they are 8 cysteines.

with two listed disulfide bonds (C233-C240) and (C791-C851) and lone cysteines C268 and C531; C531 seems to have a problem in that loop? Here again C871 is close enough to C909 to form a disulfide bond?

In RhopH3 (Chain C in the PDB file) they are 14 cysteines.

with lone cysteines C336 and C446 and two listed disulfide bonds: (C157-C231); (C244-C253). That said, pairs (C42//C99) (C262-?-C276) (C421-?-C620) and (C475//C536) are also close enough to form disulfide bonds?

Is this a definitive assignment or are further adjustments needed/expected?

2) TMDs in RhopH complex.

Authors describe precisely the TMD in CLAG3 (α44) as potentially forming the trans-membrane pore in PSAC (where they previously mapped the mutation A1215) but they don't define explicitly the predicted TMDs in RhopH2 and RhopH3 in the manuscript. They are drawn on Figure 3G and Figure 3—figure supplement 1 panels E and F and one can roughly guess their position from the Phobius plots in panels C and D. It took me a while to figure out where to look in the structure.

As pointed earlier, it is a very interesting structure but it is not always well described. Since this is a new structure describing 3 novel proteins from *Plasmodium* (with no real structural homologues available), a potential membrane protein (complex) it is not like that they are legions. I count two so far, it is worth the effort to give a complete classical diagram for each protein with secondary structure elements on top of the sequence and highlighting some of their most salient features on it (at least the TMDs, the HVR, the interactions surfaces). I understand that we cryo-EM and X-ray people are solving structures of increasing complexity, faster than we can write and maybe think but this should not prevent us from describing them with a minimum of precision so it makes the work of the reviewer and potential readers less painful and more interesting.

Using DALI, similarities with BCl^-^xL, SepL and γ-secretase APH-1 for TMD portions of RhopH2, H3 and CLAG3, respectively to support their claims.

Figure 3—figure supplement 1B. Despite its legend and knowing what Colicin and Bxl are and look like, I have absolutely no clue what that panel actually shows? It is not clearly illustrated.

3) “Interaction with PTEX” for trafficking across the PVM.

In Figure 4, the Authors demonstrate the existence of two pools but I am not convinced that the Authors provide strong evidence that PTEX is interacting with RhopH for its translocation across the PVM. One could argue that PTEX knockdown results in the lack of export of another protein that is directly or indirectly required for the subsequent proper maturation of RhopH from its soluble to integral form (PSAC) at the RBCM while RhopH follows another route for translocation across the PVM (not through the PTEX, not HSP101-dependent)? That point of the manuscript is rather weak and remains contentious as the Authors mention contradicting reports from Beck et al., 2014, and their own work from Ito et al., 2017. Should RhopH proteins really transit through PTEX then they are PEXEL-negative proteins.

Although the model is attractive, it is hard for me to understand how a ternary complex so tightly assembled with so many disulfide bonds would be threaded through the membrane via PTEX to be refolded and reassembled on the other side (whether in it is in the same soluble form or directly into a membrane-inserted integral form). All three TMDs predicted in each subunit are in the middle of each protein. Integration/insertion in the host cell membrane will then not only require considerable conformational rearrangement but also the translocation of protein sub-domains on opposite sides of the bilayer. While this is discussed where authors mentioned the absence of a lateral gate in the PTEX PVM pore subunit EXP2 for lateral insertion of TM segments, they do not rule out the intervention of other chaperones or other trafficking pathways: like vesicular trafficking maybe via tubular network extensions/Maurer's clefts. That latter route seems more likely.

Nevertheless, the Authors certainly draw 1) an interesting model for the “life cycle” of the RhopH complex and 2) a parallel between their system and the smaller pore forming toxins that also transition from a soluble monomeric form to an oligomeric membrane-inserted pore. However, their system is heterotrimeric thus suggesting a singular increase in complexity compared to the majority of single- or bi-component PFTs. What would be the receptor (usually a lipid in PFTs) triggering insertion inside the host cell membrane?

It is unfortunate that the Authors cannot visualize insertion (if it is spontaneous like PFTs and does not require extra energy) on membranes using their purified endogenous soluble complex. I realize they are limited by the amount of material available.

The Figure 5 conceptualizing their model is nice but somehow confusing.I suggest labeling the PV and/or PVM, the parasite, the RBC for clarification. Although the numbers give a false impression of ordered steps. While 1 and 2 are connected, I would think that here 3 does not follow 2. 3 is de novo endogenous synthesis of RhopH from a parasite dividing inside the infected red blood cell. And while PSAC in 4 might be the result of translocation through PTEX as the Authors propose in this manuscript, it could also be the result of another insertion/secretion path through tubular network maybe. Is there a reason why the inserted form of CLAG3 (PSAC) has two TM spanning segments drawn? Is it confirmed that both Rhop2 and Rhop3 are on the cytoplasmic side of the erythrocyte in the inserted form?

Reviewer #4:

Please note that I can only judge the cell biological and biochemical aspects of the manuscript. The authors investigate the dual soluble/membrane integral nature of the RhopH complex described previously, showing that it is soluble at late stages (in schizonts) and transforms into a membrane-integral form after invasion and that PTEX activity is required for this transition. The work is well done and the data are clear and convincing. The structures show that the TM are folded into interior of the protein, providing a convincing reason why the complex is initially soluble. However, there is no indication how the transition from soluble to membrane-bound may occur may occur and the data implicating the PTEX in transport of the complex are very indirect and open to other interpretations.

1) The authors claim that the PTEX is involved in transport of the RhopH complex to the host cell, but do not provide evidence for this. The experiment in Figure 4E shows that PTEX is required for insertion of the complex into the membrane, but does not prove any direct interaction of the PTEX with the complex. There are several other explanations for the finding that PTEX activity is required for membrane insertion of the complex, such as the requirement of an exported accessory factor.

[Editors’ note: further revisions were suggested prior to acceptance, as described below.]

Thank you for submitting your article "Malaria parasites use a soluble RhopH complex for erythrocyte invasion and an integral form for nutrient uptake" for consideration by *eLife*. Your article has been reviewed by four peer reviewers, one of whom is a member of our Board of Reviewing Editors, and the evaluation has been overseen by Dominique Soldati-Favre as the Senior Editor. The reviewers have opted to remain anonymous.

The reviewers have discussed the reviews with one another and the Reviewing Editor has drafted this decision to help you prepare a revised submission.

Summary:

This is an interesting study of *Plasmodium falciparum* RhopH, a complex of three proteins (CLAG, RhopH2 and RhopH3) that plays a dual role in the malaria parasite, first in merozoites for invasion of erythrocytes, then at the erythrocyte membrane for nutrient uptake by the parasite. The authors developed a freeze-thaw procedure for gentle detergent-free harvest of a soluble form of the RhopH complex, and used cryo-EM to determine de novo the structure of the complex at high resolution. Interestingly, predicted transmembrane domains are shielded in the soluble complex, suggesting a model where the complex is transferred to the erythrocyte as a soluble form, and then undergoes large-scale conformational changes for insertion in the host erythrocyte membrane. The authors then conducted a thorough biochemical characterization of RhopH and show that the complex is synthesized as a non-integral complex, which is transferred into the infected erythrocyte during merozoite invasion. The complex is then exported to the host cell membrane, a process that possibly depends on the PTEX protein translocon.

Revisions:

The four reviewers agree that this work warrants publication in *eLife* and that the revised manuscript would be suitable without additional experimental data. However, they also pointed at remaining issues of clarity and presentation that need to be addressed. The individual comments are pasted below.

Reviewer #1:

The authors resubmit an improved version of their manuscript. In particular, they describe the structure of the RhopH complex with more details, and added new data concerning the biochemical characterization of the complex. In particular, analysis of purified merozoites confirms that only the peripheral form of CLAG is found at this stage, corroborating the proposed model for RhopH synthesis and export. The authors partially clarified in the revised text the nature of the peripheral (carbonate-extractable) and integral (carbonate-resistant) protein pools. However, some aspects in the fractionation studies still need to be clarified, and the role of the PTEX translocon should be considered with more caution.

– There is still a lack of clarity regarding the soluble form in the fractionation studies. In the Materials and methods, the authors indicate that after parasite resuspension in lysis buffer (7.5 mM Na2HPO4, 1 mM EDTA, pH 7.5), followed by ultracentrifugation, the supernatant was kept as the "soluble" fraction. CLAG3 is detected in this soluble fraction, as shown in Figure 4A, 4D, 4F. It is unclear what this soluble form corresponds to. Is it the same fraction as indicated as "freeze-thaw" in Figure 4G? If not, why is CLAG3 released in the supernatant upon parasite treatment with the buffer alone? This contradicts the conclusions drawn from Figure 1B, stating that hypotonic lysis is not sufficient to release the peripheral complex.

– Subsection “RhopH is synthesized as a non-integral complex”: the pronase experiment (Figure 4A-B) shows that the integral pool is exposed on the cell surface, but is not informative on the timing of synthesis.

– In Figure 4C, an additional loading control should be included to probe a merozoite integral membrane protein, similarly to band3 used for schizonts.

– In Figure 4F, why is there an increase in the soluble form upon removal of TMP? In Figure 4G, why is there a reduction in the carbonate-extracted pool after protease treatment?

– The authors acknowledge that there is no evidence for a direct role of the PTEX translocon in export, and now refer to a "PTEX translocon-dependent" export mechanism. However, PTEX conditional knockdown parasites are unable to progress through development and arrest at the ring stage. The authors cannot exclude that the observed defect in the switch from soluble to integral protein complex is due to the developmental arrest of the 13F10 mutant, possibly before the expression (or activation) of a translocon-independent mechanism for export, which normally takes place later during parasite development. This limitation should be mentioned in the Discussion, and an alternative pathway (arrow) with a question mark could be added in the model in Figure 5.

Reviewer #2:

The manuscript has been clearly improved and the Discussion is also strong to describe and propose this exciting structure and model. I support publication.

Concerning the structure.

Thank you for Figures 2D, 2G and H. It helps (which is nice) and I sense it is more accurate now than it was before (which is desirable).

The disulfide issue has been addressed. I am not particularly fond of disulfide bonds but since these are secreted proteins, with no obvious structural homologues, I think it was worth checking them as it could be of use to others in the future. I am happy to see that now 4 observed disulfide bonds have appeared (in addition to the first 7 described) in a decent model, a sign that the model builder(s) took the time and effort to build a structure as good as possible, that reflects the data and that was thoroughly looked at by its “owners” and not only by this reviewer. Thank you for providing us with a complete cryo-EM table including PDB and EMDB codes.

It seems now to pass quality control for an overall 2.9 A resolution structure. Thank you!

Thank you for clarifying Figure 5 to me and strengthening and improving the discussion on the model for RhopH with Figures 4C and 4G. I got it now.

I really look forward to see the membrane-inserted form. This will also be a beautiful story.

Reviewer #3:

The authors do a very good job addressing the reviewers' comments on the manuscript. Especially the addition of the investigation of the merozoites and the fractionation experiment in Figures 4C and 4G, respectively, provide clear insight into the localization of the soluble and membrane-bound fractions and the timing of the transition between the two states. Scientifically, I have no more comments. However, there are a few minor issues that may lead to confusion on the part of the reader and although the authors did an admirable job adjusting the language describing the model, some of the new data do not seem to be reflected in the model.

Subsection “Subunit interactions and roles”: The description of RhopH2 and BCl^-^xL and RhopH3 and SepL, respectively, as orthologues overstates the amount of similarity; orthologues are defined having identical function and potentially the ability to complement each other's function. The authors correctly state that they performed a similarity search, so it would be more correct to state that these proteins contain regions of structural similarity, rather than refer to them as homologues.

Discussion paragraph one: the first half of the sentence "Alternatively, monoclonal antibody…" is unclear.

Discussion paragraph two: the phrase "failed transit and membrane insertion in the PTEX knockdown (Figure 4F) may, nonetheless, be an indirect result of blocked export of multiple effector proteins" is only partially supported by the results. The authors very convincingly show that the complex can be present in a soluble (non-membrane-inserted) state and that the complex is found in this state in parasites in which PTEX synthesis has been knocked down. However, no evidence for the localization of RhopH in these parasites is provided and hence no conclusion about failed transit of the complex (assuming that the authors refer to the passage of the PVM) in these parasites can be drawn.

The sentence "Upon erythrocyte invasion, these and other rhoptry proteins are deposited into the parasitophorous vacuole, where the PTEX protein translocon mediates export into host cytosol (de Koning-Ward et al., 2009; Beck et al., 2014; Ho et al., 2018)." This sentence suggests that the PTEX transports rhoptry proteins across the PVM. To my knowledge, there are no data showing that rhoptry protein is transported by the PTEX; RhopH would be the first, if the role of PTEX in the transit of the complex past the PV is confirmed. Perhaps this can be rephrased to remove the suggested (although not explicitly stated) link between deposition of rhoptry proteins into the PV and protein export by the PTEX?

Figure 4D. The experiment showing the membrane association of the complex at different stages in the erythrocytic lifecycle is a valuable addition to the manuscript. It is unclear how the results fit into the model that the authors present, however. RhopH is deposited into the PV upon invasion and PTEX-dependent protein export starts at most minutes later. It is thus expected that RhopH is transported to the cytosol of the host cell almost immediately after invasion. As the complex remains in a soluble state for an extended time after this, it seems unlikely that PTEX is directly responsible for the membrane insertion of CLAG. The results rather seem to support a model in which an accessory factor, produced during the trophozoite stage and exported through the PTEX, is responsible for the transition from a soluble to a membrane-bound state.

Reviewer #4:

Further comments relating to the revised text are as follows:

– In Figure 1, I find the labelling of the timing of each step confusing, as the timing is relative to Rhoptry complex synthesis instead of from invasion

– In Figure 4D, the soluble, CO3 and membrane bands to do not seem to "add up" to what is observed in the total for Ring material, has some material been lost?

– In the same figure, the most amount of CO3 material is observed in schizont material, does this represent the newly synthesised material? Can this be made more clear in the text

– In Figure 4G, what stage are the parasites at?

– In Figure 4G, what is the purpose of adding the adolase panel?

If these comments are addressed I would recommend this paper be published in *eLife*.

---

## [Author Response]

[Editors’ note: the authors resubmitted a revised version of the paper for consideration. What follows is the authors’ response to the first round of review.]

Reviewer #1:[…]1) Cryo-EM as used to determine the structure of a soluble form of RhopH, which apparently corresponds to a minor fraction of the complex "not associated with membranes". It is not clear what exactly this fraction corresponds to. In a previous report (Ito et al., 2017) the same group indicated that members of the complex are membrane-associated, but extractable in part by carbonate. Why is this fraction not recovered after hypotonic lysis?It is not clear whether the soluble form corresponds to peripheral membrane proteins.Have the authors attempted recovering the complex from purified merozoites or spent media for cryoEM? The authors propose that a change in conformation explains the differential behavior of the complex, but they should discuss alternative explanations, such as post-translational modifications or protein interactions, which could alter the association of the complex with the membranes.

We apologize for the confusion created by the phrase “not associated with membranes”. We intended to indicate that the CLAG3 released by freeze-thaw is no longer associated with membranes. Our present findings remain consistent with quantitative association of CLAG3 with membranes, first as a peripheral protein after synthesis and much later as an integral protein at the host membrane. We have revised this phrase to clarify and to emphasize that the freeze-thaw released protein is peripheral (“some CLAG3 from the peripheral pool”).

It is correct that hypotonic lysis liberates negligible amounts of RhopH proteins at any parasite stage, establishing that the RhopH complex is constitutively associated with membranes in either peripheral or integral pools. Partial release with simple freeze-thaw, as we now report for the first time, implicates loose membrane association for the peripheral pool, at least at some parasite stages. Alkaline carbonate treatment is harsher (pH 11) and is the most broadly accepted test for distinguishing between peripheral and integral proteins. Consistent with this rank order of stringency, we found that freeze-thaw releases smaller amounts of peripheral RhopH than alkaline carbonate treatment (Figure 1B). Because carbonate extraction is harsh and denatures proteins, identification of gentle, non-denaturing release with freeze-thaw proved enabling for structure determination. We have revised the manuscript to clarify this (“although Na_2_CO_3 [_…] many proteins”).

Our manuscript includes a 3D low resolution structure from spent medium, using negative stain imaging (original Figure 1G). Because it is similar to the freeze-thaw reconstruction in Figure 1G, we could not commit resources and manpower to perform cryo-EM imaging of the essentially identical complex from spent medium.

We agree that post-translational modifications may facilitate the conformational changes needed for CLAG3 membrane insertion and have revised the Discussion accordingly (“these rearrangements […] acetylation”). Please note, however, that post-translational modifications should not be viewed as an “alternate explanation” because the CLAG3 α-helix 44 is buried in the soluble complex and must undergo reorientation for membrane insertion regardless of subsequent modifications.

2) The biochemical characterization of the different pools of RhopH complex in schizonts is complicated by the coexistence of previously synthesized RhopH (integral form) and newly synthesized protein (peripheral, in the rhoptries). In Figure 4, the authors should include purified merozoites, which should contain only the peripheral/soluble complex, and erythrocyte ghost membranes, which should only contain the integral form. This would provide clearer evidence to support their model.

We agree with these comments and have now performed fractionation studies with freed merozoites, as requested. The new Figure 4C shows that freed merozoites contain CLAG3 in only a peripheral form, in contrast to their corresponding schizont progenitor. We include Band3 as an endogenous erythrocyte membrane protein to confirm that the freed merozoites are not contaminated with RBC membranes and to further establish that carbonate extraction reveals both peripheral and integral CLAG3 pools in schizonts but only an integral pool for Band3, a more conventional membrane protein. We agree with this reviewer that this result provides independent evidence for the proposed model. Please see the revised Results, “Fractionation studies using purified merozoites …”, and corresponding figure legend.

Unfortunately, there is not a good procedure for harvesting erythrocyte ghost membranes from infected cells. A commonly discussed protocol (PMID: 2001227) uses host membrane lysis with saponin, mechanical shearing, and differential centrifugation; their paper used membrane-specific markers to assess harvest and contamination. While their data are useful, our unpublished studies with this procedure (previously and during the present work) have produced low recovery and likely contamination with Maurer’s clefts (a problem not evaluated in their study).

To provide analogous evidence for the model, we instead performed protease susceptibility studies with the epitope-tagged parasite used in our cryo-EM studies. The revised Figure 4G shows that while freeze-thaw released and carbonate-extractable CLAG3 are not cleaved by extracellular protease, the integral pool at the host membrane is susceptible. Notably, the observed c-terminal cleavage product is also carbonate-resistant, consistent with α-helix 44 serving as a transmembrane domain. Importantly, this finding indicates that host membrane CLAG3 is indeed integral. Please see Figure 4G, the revised Results “Stage-dependent membrane […] at the host membrane.” and revised Figure 4 legend.

These findings compellingly show that CLAG3 is synthesized and transferred to new erythrocytes as a soluble protein and that it subsequently becomes integral at the host membrane to become exposed to host plasma.

3) In the PTEX mutant experiment, it is crucial to analyze parasites at the same stage. Beck et al., 2014, showed that in the absence of TMP, 13F10 mutant parasites are blocked at the ring stage, so it is not surprising that the protein extraction profile in Figure 4E is similar to the ring profile in Figure 4C, and does not prove that export through PTEX is required for RhopH membrane insertion. This is a major weakness in the model.

PTEX conditional knockdown parasites are indeed unable to progress through development and resemble rings in their morphology. This concern, an indirect effect of PTEX block, applies equally well to other exported parasite proteins (e.g. PfEMP1 in the original 13F10 mutant studies, PMID: 25043010, where the authors stated “although […] delivery of PfEMP1 to the RBC surface is HSP101-dependent, we cannot exclude the possibility that this block is an indirect result of a failure to export other proteins required for PfEMP1…”).

The evidence for RhopH is better than for PfEMP1 and some other proteins because co-IP experiments reveal that PTEX components immunoprecipitate RhopH2 (Table S4 of PMID: 19536257) and reverse co-IP using RhopH2 as the bait pulls down PTEX components (Figure 3B of PMID: 28252383).

The new data we added to the revised manuscript (question #2 above), establish that CLAG3 is synthesized as soluble protein (new Figure 4C), remains soluble upon deposit into the ring-stage parasitophorous vacuole upon invasion (Figure 4D-E), and later becomes integral at the host membrane (new Figure 4G). This timeline for membrane insertion is also suggestive of a PTEX role.

We agree that the exact mechanism of PVM translocation remains unclear; indeed, the RhopH complex may face a unique dilemma if it requires unfolding for translocation and subsequent refolding in the RBC compartment. We also agree that CLAG3 membrane insertion may occur either through direct PTEX interaction or indirectly through the action of other exported proteins.

We have significantly revised the Discussion section to present our PTEX mutant findings more cautiously, to include citations for the forward and reverse pull-down experiments that support RhopH-PTEX interaction, and to more clearly describe the intriguing questions raised by our structural and biochemical studies. We have also revised the Abstract and Introduction sections to clarify this uncertainty. We have also changed Figure 5 to represent PTEX-dependent membrane insertion more cautiously.

Reviewer #3:[…]Abstract. Although the Abstract is clear it should introduce the names of the two other rhoptry proteins RhopH2 and RhopH3 proteins simultaneously within that third sentence for the sake of clarity and consistency, as it sounds odd to bring all the attention on CLAG3 (or RhopH1) but not mention the other two subunits so intimately associated with CLAG3.

Agreed and corrected.

Second paragraph in Introduction. I would clarify “expansion in humans and vertebrates” as in “Plasmodium spp. infecting humans and other vertebrates such as birds, rodents and primates” as they explain more clearly later in the manuscript.

Agreed and corrected.

At the end of the Introduction (and maybe also in the Abstract) I would suggest being a bit more explicit in introducing PTEX (I am aware that referencing the initial study from DeKoning-Ward et al. is done much later in the text) and its function as a vacuolar translocon. In general the Introduction assumes that most readers would be familiar with the exquisitely complicated life cycle of apicomplexan parasites such as Plasmodium (or Toxoplasma): Introducing the existence and formation of a host-derived parasitophorous vacuole (membrane) following invasion of the host red blood cell by the parasite sounds necessary to clarify the general picture. Again this is done quite late in the body of the manuscript.

This is an important suggestion, which we have accepted with revisions to both the Abstract (“After transfer…”) and last paragraph of the Introduction “The complex remains […] nutrients.”

1) Structure determinationEnd of Introduction. Results section and in general. Concerning the structure itself presented at 2.9Å resolution as per cryo-EM standards. 2.9 Å is not high resolution: atomic resolution should suffice.

Actually, there is an ongoing debate about “high”, “atomic”, and “near atomic” resolution in protein structures, as discussed in “Atomic resolution: a badly abused term in structural biology” (PMID: 28375149) and the provocative responses of 5 experts and the IUCr Commission on Biological Macromolecules (PMID: 28375150). The “Sheldrick criterion” of resolution *d_min_* of 1.2 Å for “atomic” resolution may be the most conservative definition, with some respondents taking a more generous view. We agree with the consensus of these respondents that 2.9 Å, as we achieved, does not meet the standard of “atomic” or “near-atomic”. We have removed two instances of “high-resolution” that do not contribute meaning to our manuscript, but have chosen to keep two instances as they recognize that our resolution is higher than those achieved in all other cryo-EM studies of malaria parasite non-ribosomal proteins published to date.

Table 1. Statistics for cryo-EM structural analysis.In Data collection and processing. Any reason why defocus, FSC threshold and map resolution values are not indicated?In Refinement. Any reason why model resolution range is not indicated?

We have added the missing parameters to Table 1 and further improved refinement in the process.

Going through the PDB preliminary validation report indicates the presence of a fair amount of clashes in this large structure. RhopH2 in particular seems less well defined.

RhopH2 is indeed less well-defined than the other subunits. Our initial refinements permitted building of the region of RhopH2 that binds CLAG3, but multi-body refinement further improved RhopH2 density for model building, as we highlighted in the Figure 2—figure supplements 1 and 2. Nevertheless, RhopH2 remains incompletely built, possibly because of mobile domains that may serve yet-unknown functions.

Upon inspection of the structure using the provided PDB file, I will comment about disulfide bonds and "free" cysteines (reduced). The three proteins contain a fair amount of cysteines and including a total of 7 clearly identified disulfide bonds (overall the 3 subunits) as one could expect in "secreted" proteins facing harsher conditions and as the Authors say "suited for trafficking and transfer to a new host erythrocyte". These are fairly complex patterns of disulfide linkages that did not seem to catch the attention of the Authors.In CLAG3 (chain A in the PDB file) they are 14 cysteines.with lone cysteines C779 C1065 C1217 C1431and three listed disulfide bonds (C409-C415) (C523-C544) and (C1352-C1355)but pairs C335-?-C363 and C519-?-C547 seem also close enough to form disulfide bonds.In RhopH2 (chain B in the PDB file) they are 8 cysteines.with two listed disulfide bonds (C233-C240) and (C791-C851) and lone cysteines C268 and C531; C531 seems to have a problem in that loop? Here again C871 is close enough to C909 to form a disulfide bond?In RhopH3 (Chain C in the PDB file) they are 14 cysteines.with lone cysteines C336 and C446 and two listed disulfide bonds: (C157-C231); (C244-C253). That said, pairs (C42//C99) (C262-?-C276) (C421-?-C620) and (C475//C536) are also close enough to form disulfide bonds?Is this a definitive assignment or are further adjustments needed/expected?

Yes, our original submission failed to mention the numerous cysteines in the three subunits, as recognized by early workers (PMID: 15953647). We have checked the maps (confirming nearly all this reviewer’s careful tallies above) and added both an orienting ribbon schematic and a tabulated list (new Figure 2G and H). We have also revised the Results (“Each subunit has numerous…”), Discussion (“If it transits directly….”) and Abstract (“tightly assembled with extensive disulfide bonding”) to highlight this important feature.

2) TMDs in RhopH complex.Authors describe precisely the TMD in CLAG3 (α44) as potentially forming the trans-membrane pore in PSAC (where they previously mapped the mutation A1215) but they don't define explicitly the predicted TMDs in RhopH2 and RhopH3 in the manuscript. They are drawn on Figure 3G and Figure 3—figure supplement 1 panels E and F and one can roughly guess their position from the Phobius plots in panels C and D. It took me a while to figure out where to look in the structure.

We have explicitly defined the positions of these helices in the text “helices defined by V740-D757 and G595-Y622 of these subunits, respectively”.

As pointed earlier, it is a very interesting structure but it is not always well described. Since this is a new structure describing 3 novel proteins from Plasmodium (with no real structural homologues available), a potential membrane protein (complex) it is not like that they are legions. I count two so far, .it is worth the effort to give a complete classical diagram for each protein with secondary structure elements on top of the sequence and highlighting some of their most salient features on it (at least the TMDs, the HVR, the interactions surfaces). I understand that we cryo-EM and X-ray people are solving structures of increasing complexity, faster than we can write and maybe think but this should not prevent us from describing them with a minimum of precision so it makes the work of the reviewer and potential readers less painful and more interesting.

A full length sequence diagram with features highlighted is not practical as such a representation for the ~3700 total residues in the complex would take many pages to represent a relatively small number of features. We hope the reviewer agrees that the new ribbon diagrams (Figure 2D and 2G) serve this purpose well and concisely.

Using DALI, similarities with BCl^-^xL, SepL and γ-secretase APH-1 for TMD portions of RhopH2, H3 and CLAG3, respectively to support their claims.Figure 3—figure supplement 1B. Despite its legend and knowing what Colicin and Bxl are and look like, I have absolutely no clue what that panel actually shows? It is not clearly illustrated.

The goal of panel B in Figure 3—figure supplement 1 is to show that some other pore-forming proteins also have soluble forms with their TMs buried. We show zoom-in structures of the relevant domains of colicin Ia and Bax with their pre-membrane inserted hydrophobic helices in green. To clarify this point for readers, we have revised this graphic to include a similar zoom-in view of CLAG3 and revised the figure legend, “(B) Cylinder view diagrams of CLAG3 […] insertion.”

3) “Interaction with PTEX” for trafficking across the PVM.In Figure 4, the Authors demonstrate the existence of two pools but I am not convinced that the Authors provide strong evidence that PTEX is interacting with RhopH for its translocation across the PVM. One could argue that PTEX knockdown results in the lack of export of another protein that is directly or indirectly required for the subsequent proper maturation of RhopH from its soluble to integral form (PSAC) at the RBCM while RhopH follows another route for translocation across the PVM (not through the PTEX, not HSP101-dependent)? That point of the manuscript is rather weak and remains contentious as the Authors mention contradicting reports from Beck et al., 2014, and their own work from Ito et al., 2017. Should RhopH proteins really transit through PTEX then they are PEXEL-negative proteins.

We agree that PTEX-dependent membrane insertion of CLAG3 may reflect either direct interaction and translocation through PTEX or an indirect mechanism such CLAG3 membrane insertion though the action of chaperone(s) that are exported via PTEX. Previous publications have used forward and reverse pull-down experiments to suggest direct interaction between RhopH2 and PTEX components (Table S4 of PMID: 19536257 and Figure 3B of PMID: 28252383).

Most importantly, we have significantly revised the Discussion section to present our PTEX mutant findings more cautiously and to more clearly describe the intriguing questions raised by our structural and biochemical studies. We have also revised the Abstract and Introduction sections to clarify this uncertainty. Finally, we added new experimental data (Figure 4C and 4G) to more strongly support the model of a soluble complex delivered to new host erythrocytes and eventually inserted at the host membrane.

Although the Beck et al., 2014 and Ito et al., 2017 papers have different IFA results for CLAG3 export into host erythrocyte cytosol, they both report that PSAC activity is abolished in the PTEX knockdown (see Figure 3A-B of the Beck paper PMID: 25043010). We understand the Beck group also confirmed that CLAG3 fails to insert in the host membrane in the knockdown, based on communications with their authors. Thus, these studies are not contradictory for the point being made in Figure 4, namely CLAG3 membrane insertion through a PTEX-dependent process. We have revised the Results to clarify this, “While two studies have obtained conflicting results about whether RhopH proteins are exported via this translocon, both reported that PTEX knockdown abolishes activation of PSAC-mediated nutrient uptake at the host membrane (Beck et al., 2014; Ito et al., 2017)”.

Although the model is attractive, it is hard for me to understand how a ternary complex so tightly assembled with so many disulfide bonds would be threaded through the membrane via PTEX to be refolded and reassembled on the other side (whether in it is in the same soluble form or directly into a membrane-inserted integral form). All three TMDs predicted in each subunit are in the middle of each protein. Integration/insertion in the host cell membrane will then not only require considerable conformational rearrangement but also the translocation of protein sub-domains on opposite sides of the bilayer. While this is discussed where authors mentioned the absence of a lateral gate in the PTEX PVM pore subunit EXP2 for lateral insertion of TM segments, they do not rule out the intervention of other chaperones or other trafficking pathways: like vesicular trafficking maybe via tubular network extensions/Maurer's clefts. That latter route seems more likely.

We agree with this excellent assessment. We have revised the Discussion to highlight this intriguing problem.

Nevertheless, the Authors certainly draw 1) an interesting model for the “life cycle” of the RhopH complex and 2) a parallel between their system and the smaller pore forming toxins that also transition from a soluble monomeric form to an oligomeric membrane-inserted pore. However, their system is heterotrimeric thus suggesting a singular increase in complexity compared to the majority of single- or bi-component PFTs. What would be the receptor (usually a lipid in PFTs) triggering insertion inside the host cell membrane?It is unfortunate that the Authors cannot visualize insertion (if it is spontaneous like PFTs and does not require extra energy) on membranes using their purified endogenous soluble complex. I realize they are limited by the amount of material available.

Another excellent point. The amount of purified RhopH complex is actually not limiting for functional reconstitution experiments, which were pursued using several distinct but unsuccessful avenues during the course of this work (esp. reconstitution of the soluble complex into planar lipid bilayers for electrical recordings of PSAC activity). The failure of these experiments may reflect involvement of chaperones in the membrane insertion process, but there are other explanations also.

The Figure 5 conceptualizing their model is nice but somehow confusing.I suggest labeling the PV and/or PVM, the parasite, the RBC for clarification. Although the numbers give a false impression of ordered steps. While 1 and 2 are connected, I would think that here 3 does not follow 2. 3 is de novo endogenous synthesis of RhopH from a parasite dividing inside the infected red blood cell. And while PSAC in 4 might be the result of translocation through PTEX as the Authors propose in this manuscript, it could also be the result of another insertion/secretion path through tubular network maybe. Is there a reason why the inserted form of CLAG3 (PSAC) has two TM spanning segments drawn? Is it confirmed that both Rhop2 and Rhop3 are on the cytoplasmic side of the erythrocyte in the inserted form?

We have revised the graphic to add key labels.

In fact, #3 does follow #2 because it is the rhoptry protein pool delivered into the PV during invasion that must then be exported into host cytosol; there is no detectable transcription/translation of RhopH proteins at ring or early trophozoite stages as would be required for de novo endogenous synthesis by the intracellular parasite.

We revised the PTEX interaction step because we agree that the original graphic too strongly suggests direct threading of RhopH proteins through PTEX. While the complex may indeed diffuse along the tubular network, the topological problem of membrane translocation would remain. Our evidence supports translocation and membrane insertion in a PTEX-dependent manner but cannot distinguish between direct or indirect mechanisms.

Prior studies have established that only a small variable region of CLAG3 (HVR in the manuscript) is surface exposed, so a second TM upstream of the HVR and helix 44 is necessary. RhopH2 and RhopH3 are not susceptible to extracellular protease, but remain associated with CLAG3 at the host membrane based on FRET studies, so are drawn as cytoplasmic. We have revised the figure legend and the corresponding Discussion section to clarify these points.

Reviewer #4:Please note that I can only judge the cell biological and biochemical aspects of the manuscript. The authors investigate the dual soluble/membrane integral nature of the RhopH complex described previously, showing that it is soluble at late stages (in schizonts) and transforms into a membrane-integral form after invasion and that PTEX activity is required for this transition. The work is well done and the data are clear and convincing. The structures show that the TM are folded into interior of the protein, providing a convincing reason why the complex is initially soluble. However, there is no indication how the transition from soluble to membrane-bound may occur may occur and the data implicating the PTEX in transport of the complex are very indirect and open to other interpretations.1) The authors claim that the PTEX is involved in transport of the RhopH complex to the host cell, but do not provide evidence for this. The experiment in Figure 4E shows that PTEX is required for insertion of the complex into the membrane, but does not prove any direct interaction of the PTEX with the complex. There are several other explanations for the finding that PTEX activity is required for membrane insertion of the complex, such as the requirement of an exported accessory factor.

We agree that PTEX-dependent membrane insertion of CLAG3 may reflect either direct interaction and translocation through PTEX or an indirect mechanism such CLAG3 membrane insertion though the action of chaperone(s) that are exported via PTEX. Previous publications have used forward and reverse pull-down experiments to suggest direct interaction between RhopH2 and PTEX components (Table S4 of PMID: 19536257 and Figure 3B of PMID: 28252383).

We have significantly revised the Discussion section to present our PTEX mutant findings more cautiously, to include citations for forward and reverse pull-down experiments that support RhopH-PTEX interaction, and to more clearly describe the intriguing new questions raised by our structural and biochemical studies. We have also revised the Abstract and Introduction sections to clarify this uncertainty. We have also added new experiments (Figure 4C and 4G) that delimit the timing of RhopH membrane insertion, supporting PTEX-dependence. Finally, we changed Figure 5 to represent PTEX-dependent membrane insertion more cautiously.

[Editors’ note: what follows is the authors’ response to the second round of review.]

Reviewer #1:The authors resubmit an improved version of their manuscript. In particular, they describe the structure of the RhopH complex with more details, and added new data concerning the biochemical characterization of the complex. In particular, analysis of purified merozoites confirms that only the peripheral form of CLAG is found at this stage, corroborating the proposed model for RhopH synthesis and export. The authors partially clarified in the revised text the nature of the peripheral (carbonate-extractable) and integral (carbonate-resistant) protein pools. However, some aspects in the fractionation studies still need to be clarified, and the role of the PTEX translocon should be considered with more caution.– There is still a lack of clarity regarding the soluble form in the fractionation studies. In the Materials and methods, the authors indicate that after parasite resuspension in lysis buffer (7.5 mM Na2HPO4, 1 mM EDTA, pH 7.5), followed by ultracentrifugation, the supernatant was kept as the "soluble" fraction. CLAG3 is detected in this soluble fraction, as shown in Figure 4A, 4D, 4F. It is unclear what this soluble form corresponds to. Is it the same fraction as indicated as "freeze-thaw" in Figure 4G? If not, why is CLAG3 released in the supernatant upon parasite treatment with the buffer alone? This contradicts the conclusions drawn from Figure 1B, stating that hypotonic lysis is not sufficient to release the peripheral complex.

Figure 4A and 4B have been corrected. The “soluble” label in these panels have been changed to “freeze-thaw” as these samples were frozen before thawing for membrane fractionation.

The experiments shown Figure 4D did not use frozen samples; a “soluble” band is detected, primarily in ring-infected cells. We suspect this relates to differences in harvest of these stages. While schizont- and trophozoite-infected cells were enriched by the percoll-sorbitol method, ring-infected cells cannot be similarly enriched. We therefore harvested and performed fractionation on rings without enrichment. Thus, a somewhat more complicated fractionation procedure with unenriched rings may account for detectable amounts of CLAG3 in panel 4D, ring stage parasites. We clarify this point in the figure legend. “While schizont- […] non-additive fractionation.”.

Figure 4F, where we used a conditional PTEX knockdown parasite for CLAG3 fractionation studies, is the most interesting. It is notable that PTEX knockdown (-TMP) yields a band in the soluble lane but +TMP does not. We do not understand this well, but a conservative hypothesis is that some of the non-integral CLAG3 pool is released from its peripheral association with membranes because of protein crowding after blocked export in the PTEX knockdown. We clarify this possibility in the Results: “CLAG3 that failed to insert […] export from the parasitophorous vacuole.”.

– Subsection “RhopH is synthesized as a non-integral complex”: the pronase experiment (Figure 4A-B) shows that the integral pool is exposed on the cell surface, but is not informative on the timing of synthesis.

Yes, the timing of synthesis was previously established based on stage-specific transcription and translation in schizonts. This was stated in the manuscript Introduction with citation to Ling et al., 2005.

– In Figure 4C, an additional loading control should be included to probe a merozoite integral membrane protein, similarly to band3 used for schizonts.

Figure 4C shows Band3, not as a loading control, but to demonstrate that the freed merozoites are not contaminated by erythrocyte membranes. Band3 is detected in schizont-infected erythrocytes but not in freed merozoites, indicating that these merozoites are not contaminated with erythrocyte membrane fragments. As this experiment is not designed to quantitate protein abundance in merozoites vs schizonts, we do not see value in a merozoite “loading control”. Indeed, the procedure for freed merozoite harvest sacrifices yield to obtain high purity (*PNAS* 108:13275-80).

– In Figure 4F, why is there an increase in the soluble form upon removal of TMP? In Figure 4G, why is there a reduction in the carbonate-extracted pool after protease treatment?

As described above, we clarify the increase in the soluble form upon PTEX knockdown (Figure 4F) in the Results: “CLAG3 that failed to insert […] export from the parasitophorous vacuole.”

The difference the reviewer observes between – and + protease for the “CO_3_^=^” lanes is not statistically significant (*P =* 0.42, paired Student *t* test from 4 matched trials). We do not think it is necessary to explicitly state this in the paper as it is tangential to this experiment’s purpose.

– The authors acknowledge that there is no evidence for a direct role of the PTEX translocon in export, and now refer to a "PTEX translocon-dependent" export mechanism. However, PTEX conditional knockdown parasites are unable to progress through development and arrest at the ring stage. The authors cannot exclude that the observed defect in the switch from soluble to integral protein complex is due to the developmental arrest of the 13F10 mutant, possibly before the expression (or activation) of a translocon-independent mechanism for export, which normally takes place later during parasite development. This limitation should be mentioned in the Discussion, and an alternative pathway (arrow) with a question mark could be added in the model in Figure 5.

Apologies for an intentional double negative, but we do not acknowledge that there is no evidence for a direct role in of the PTEX translocon in export. To the contrary, previous IFA experiments provide strong evidence. Please see Figure 7B of *eLife* 6:e23485 (2017), where immunofluorescence experiments show that all three RhopH components are trapped in the parasitophorous vacuole upon TMP removal in the PTEX conditional knockdown. Forward and reverse coimmunoprecipitation experiments from other groups (cited in the first paragraph) also provide evidence for direct interaction between PTEX and RhopH proteins to mediate export.

We also do not refer to a “PTEX translocon-dependent export mechanism”. The 5 instances of “translocon-dependent”, “PTEX-dependent” or “dependent on PTEX activity” are all careful to state that RhopH membrane insertion is PTEX-dependent.

These reviewer statements incorrectly conflate PTEX-dependent “export” and “membrane insertion”. These processes should be considered separately. Export has been studied previously (*eLife* 6:e23485 and Nature 511:592, where only some findings are contradictory), requires further study, but is not a question we have pursued here because it will require new technologies.

The membrane insertion event is the important question our manuscript examines for the first time, as it is enabled by our high-resolution structure. We agree with the reviewer that insertion may occur either directly during PVM translocation or later through the action of exported chaperones, and hence refer to membrane insertion as “PTEX-dependent”. This word choice is important and appropriately cautious.

We have revised the Discussion to clarify these points, “We determined that the complex […] may result from blocked export of multiple effector proteins”.

We do not agree with the reviewer suggestion to add an arrow with question mark to Figure 5 because it would incorrectly imply that there is a route for RhopH membrane insertion that is completely independent of PTEX. The data clearly indicate that there is no such route, so a graphic such as this would be misleading.

Reviewer #3:The authors do a very good job addressing the reviewers' comments on the manuscript. Especially the addition of the investigation of the merozoites and the fractionation experiment in Figures 4C and 4G, respectively, provide clear insight into the localization of the soluble and membrane-bound fractions and the timing of the transition between the two states. Scientifically, I have no more comments. However, there are a few minor issues that may lead to confusion on the part of the reader and although the authors did an admirable job adjusting the language describing the model, some of the new data do not seem to be reflected in the model.Subsection “Subunit interactions and roles”: The description of RhopH2 and BCl^-^xL and RhopH3 and SepL, respectively, as orthologues overstates the amount of similarity; orthologues are defined having identical function and potentially the ability to complement each other's function. The authors correctly state that they performed a similarity search, so it would be more correct to state that these proteins contain regions of structural similarity, rather than refer to them as homologues.

Although it is conventional to refer to protein 3D structure comparisons, as used to find BCl^-^xL and SepL, as “structural homology searches” and to call the hits “structural homologs”, we want to be abundantly cautious and address this reviewer’s point. We have therefore changed “structural homologs” to “hits from our structural similarity searches”.

Discussion paragraph one: the first half of the sentence "Alternatively, monoclonal antibody…" is unclear.

Yes, this was a bit awkward. We have revised to this sentence to clarify. “The RhopH3 C-terminus …”.

Discussion paragraph two: the phrase "failed transit and membrane insertion in the PTEX knockdown (Figure 4F) may, nonetheless, be an indirect result of blocked export of multiple effector proteins" is only partially supported by the results. The authors very convincingly show that the complex can be present in a soluble (non-membrane-inserted) state and that the complex is found in this state in parasites in which PTEX synthesis has been knocked down. However, no evidence for the localization of RhopH in these parasites is provided and hence no conclusion about failed transit of the complex (assuming that the authors refer to the passage of the PVM) in these parasites can be drawn.

There is already published data for this point. Please see Figure 7B of *eLife* 6:e23485 (2017), where immunofluorescence experiments revealed that all three RhopH components are trapped in the parasitophorous vacuole upon TMP removal in the PTEX conditional knockdown.

We have revised the Discussion to clarify that PTEX knockdown traps RhopH proteins in the parasitophorous vacuole, based on the prior IFA experiments. “We determined that the complex […] may result from blocked export of multiple effector proteins”.

The sentence "Upon erythrocyte invasion, these and other rhoptry proteins are deposited into the parasitophorous vacuole, where the PTEX protein translocon mediates export into host cytosol (de Koning-Ward et al., 2009; Beck et al., 2014; Ho et al., 2018)." This sentence suggests that the PTEX transports rhoptry proteins across the PVM. To my knowledge, there are no data showing that rhoptry protein is transported by the PTEX; RhopH would be the first, if the role of PTEX in the transit of the complex past the PV is confirmed. Perhaps this can be rephrased to remove the suggested (although not explicitly stated) link between deposition of rhoptry proteins into the PV and protein export by the PTEX?

We have revised this sentence to avoid such inference and to highlight that PTEX-mediated export of RhopH would be a first. “The PTEX protein translocon […] deposited in the vacuole”.

Figure 4D. The experiment showing the membrane association of the complex at different stages in the erythrocytic lifecycle is a valuable addition to the manuscript. It is unclear how the results fit into the model that the authors present, however. RhopH is deposited into the PV upon invasion and PTEX-dependent protein export starts at most minutes later. It is thus expected that RhopH is transported to the cytosol of the host cell almost immediately after invasion. As the complex remains in a soluble state for an extended time after this, it seems unlikely that PTEX is directly responsible for the membrane insertion of CLAG. The results rather seem to support a model in which an accessory factor, produced during the trophozoite stage and exported through the PTEX, is responsible for the transition from a soluble to a membrane-bound state.

We respectfully point out that this comment assumes that PTEX-dependent export is constitutively applied to all cargo proteins present in the vacuole. This assumption has not been tested experimentally. Because resident soluble proteins are not exported, it seems likely that there are unidentified mechanisms that control timing and targeting of proteins in the PV for export, in contradiction to this implicit assumption. We also point out that IFA experiments with PTEX knockdown parasites reveal RhopH proteins trapped in the PV. The revised the paragraph addresses this point, “We determined that the complex […] may result from blocked export of multiple effector proteins”.

Reviewer #4:Further comments relating to the revised text are as follows:– In Figure 1, I find the labelling of the timing of each step confusing, as the timing is relative to Rhoptry complex synthesis instead of from invasion

Because this manuscript tracks RhopH from its point of synthesis, we think it is best to have the graphic labelled according to this complex’s synthesis. As the biology and trafficking of this complex are complicated, we believe this will help readers. A timing based on invasion would force labeling of RhopH synthesis at 38-44 h and confuse more readers than not.

– In Figure 4D, the soluble, CO3 and membrane bands to do not seem to "add up" to what is observed in the total for Ring material, has some material been lost?

We have added a statement about “non-additive fractionation” to the Figure 4, panel D legend. This likely resulted because ring-infected cells cannot be enriched by the percoll-sorbitol method, requiring fractionation of rings at lower parasitemias than possible for schizont- and trophozoite-infected cells.

– In the same figure, the most amount of CO3 material is observed in schizont material, does this represent the newly synthesised material? Can this be made more clear in the text

Yes, this is almost certainly the newly synthesized protein, as supported by freed merozoite and protease-susceptibility studies (Figure 4C and 4A-B). We have made this more explicit: “a primarily extractable form upon synthesis in schizonts …”

– In Figure 4G, what stage are the parasites at?– In Figure 4G, what is the purpose of adding the adolase panel?

This experiment used mixed trophozoite- and schizont-infected cells, as now indicated in the figure legend, “enriched mature infected cells”. Aldolase is shown as conventional soluble protein that is quantitatively extracted by freeze-thaw and carbonate treatment. Similarly, EXP2 is shown as a more conventional integral membrane protein that is minimally extracted by these treatments. Comparison to these proteins highlights that CLAG3 is unusual, present as both soluble and integral forms within infected cells. We hope this comparison is clear from the Figure 4G legend.